# Embedding a One-column Ocean Model (SIT 1.06) in the Community Atmosphere Model 5.3 (CAM5.3; CAM5–SIT v1.0) to Improve Madden–Julian Oscillation Simulation in Boreal Winter

Yung-Yao Lan, Huang-Hsiung Hsu[*], Wan-Ling Tseng, and Li-Chiang Jiang

Research Center for Environmental Changes, Academia Sinica, Taipei 11529, Taiwan

[*]*Correspondence to*: Huang-Hsiung Hsu (hhhsu@gate.sinica.edu.tw)

**Abstract**
The effect of the air–sea interaction on the Madden–Julian Oscillation (MJO)
was investigated using the one-column ocean model Snow–Ice–Thermocline (SIT
1.06) embedded in the Community Atmosphere Model 5.3 (CAM5.3; hereafter
CAM5–SIT v1.0). The SIT model with 41 vertical layers was developed to simulate
sea surface temperature (SST) and upper-ocean temperature variations with a high
vertical resolution that resolves the cool skin and diurnal warm layer and the upper
oceanic temperature gradient. A series of 30-year sensitivity experiments were
conducted in which various model configurations (e.g., coupled versus uncoupled,
vertical resolution and depth of the SIT model, coupling domains, and absence of the
diurnal cycle) were considered to evaluate the effect of air–sea coupling on MJO
simulation. Most of the CAM5–SIT experiments exhibit higher fidelity than the
CAM5-alone experiment in characterizing the basic features of the MJO such as
spatiotemporal variability and the eastward propagation in boreal winter. The overall
MJO simulation performance of CAM5–SIT benefits from (1) better resolving the
fine vertical structure of upper-ocean temperature and therefore the air–sea interaction
that result in more realistic intraseasonal variability in both SST and atmospheric
circulation and (2) the adequate thickness of a vertically-gridded ocean layer. The
sensitivity experiments demonstrate the necessity of coupling the tropical eastern
Pacific in addition to the tropical Indian Ocean and the tropical western Pacific.
Coupling is more essential in the south than north of the equator in the tropical
western Pacific. Enhanced MJO could be obtained without considering the diurnal
cycle in coupling.

## 1. Introduction

The Madden–Julian Oscillation (MJO) is a tropical large-scale convection circulation system that propagates eastward across the warm pool region from the tropical Indian Ocean (IO) to the western Pacific (WP) on an intraseasonal time scale (Madden and Julian, 1972). The MJO is not just an atmospheric phenomenon. The findings from a multi-nation field campaign in the tropics called the Dynamics of MJO/Cooperative Indian Ocean Experiment on Intraseasonal Variability in the Year 2011 (DYNAMO/CINDY2011; de Szoeke et al., 2017; Johnson and Ciesielski, 2017; Pujiana et al., 2018; Yoneyama et al., 2013; Zhang and Yoneyama, 2017) revealed vigorous air–sea coupling during the evolution of the MJO (Chang et al., 2019; DeMott et al., 2015; Jiang et al., 2015, 2020; Kim et al., 2010; Li et al., 2016; Li et al., 2020; Newman et al., 2009; Pei et al., 2018; Tseng et al., 2014). During the suppression of convection, the MJO propagates eastward with light winds, which is accompanied by enhanced downwelling shortwave radiation absorption, weaker upward latent and sensible fluxes, less cloudiness and precipitation, and weaker vertically turbulent mixing in the upper ocean, thus causing an increase in the upper-ocean temperature. In the following active phase when deep convection occurs, downwelling shortwave radiation is reduced and stronger westerly winds enhance latent/sensible heat flux (LHF/SHF) loss from the ocean surface, thus causing a decrease in the upper-ocean temperature (DeMott et al., 2015; Madden and Julian, 1972, 1994; Zhang, 2005).

In addition to the tropical ocean surface, the structure of the upper ocean also evolves. Alappattu et al. (2017) reported that during an MJO event, surface flux perturbations cause changes in the ocean thermohaline structure, thus affecting the mixed-layer temperature. The following change in sea surface temperature (SST) can further affect atmospheric circulation of the MJO. Variations in SST mediate LHF and

SHF exchange across the air–sea interface. Although SST responds to atmospheric
forcing, the modulation of LHF and SHF provides feedback to the atmosphere
(DeMott et al., 2015; Jiang et al., 2020). Li et al. (2008, 2020) proposed that the phase
relationship between SST and convection implies a delayed air–sea interaction
mechanism whereby a preceding active-phase MJO may trigger an inactive-phase
MJO through the delayed effect of the induced SST anomaly over the IO. The
reduction in SST caused by a preceding active-phase MJO may, in turn, yield delayed
ocean feedback that initiates a suppressed-phase MJO, and vice versa. The by-no-
means negligible effect of intraseasonal SST variations caused by surface heat fluxes
suggests that the ocean state can affect the MJO (DeMott et al., 2015, 2019; Hong et
al., 2017; Li et al., 2020).

Since its discovery almost five decades ago, the MJO remains a phenomenon

that poses a challenge to the capacity of state-of-the-art atmospheric general
circulation models (AGCMs) such as those participating in the Coupled Model
Intercomparison Project phase 5 and 6 to generate successful simulations (Ahn et al.,
2017, 2020; Bui and Maloney 2018; Jiang et al., 2020; Hung et al., 2013; Kim et al.,

2011).

Recent studies have reported that air–sea coupling improves the representation of

the MJO in numerical simulation (Bernie et al., 2008; Crueger et al., 2013; DeMott et
al., 2015; Li et al., 2016; Li et al., 2020; Tseng et al., 2014; Woolnough et al., 2007).
Tseng et al. (2014) indicated that effectively resolving the tropical upper-ocean warm
layer to capture temperature variations in the upper few meters of the ocean could
improve MJO simulation. DeMott et al. (2015) suggested that the tropical
atmosphere–ocean interaction may sustain or amplify the pattern of the enhanced and
suppressed atmospheric convection of the eastward propagation. DeMott et al. (2019)
demonstrated that the improved MJO eastward propagation in four coupled models
resulted from enhanced low-level convective moistening for a rainfall rate of >5 mm
day$^{-1}$ due to air–sea coupling. In addition, numerical experiments have been
performed to investigate the effect of the diurnal cycle on the MJO (Hagos et al.,
2016; Oh et al., 2013), with the results suggesting that the strength and propagation of
the MJO through the Maritime Continent (MC) were enhanced when the diurnal cycle
was ignored.

Although previous studies have demonstrated the importance of considering the

air–sea interaction in a numerical model to improve MJO simulation, additional
details regarding model configuration (e.g., vertical resolution and total depth of the
vertically-gridded ocean, coupling domain, and absence of the diurnal cycle in air–sea
coupling) have not been systematically explored. Tseng et al. (2014) coupled the one-
column ocean model Snow–Ice–Thermocline (SIT; Tu and Tsuang, 2005) to the fifth
generation of the ECHAM AGCM (ECHAM5–SIT) in the tropics and indicated that a
vertical resolution of 1 m was essential to yield an improved simulation of the MJO
with a realistic strength and eastward propagation speed.

In this study, we coupled the SIT model to the Community Atmosphere Model

version 5.3 (CAM5.3; Neale et al., 2012)—the atmosphere component of the
Community Earth System Model version 1.2.2 (CESM1.2.2; Hurrell et al., 2013) —to
explore the improvement of MJO simulation by coupling SIT model to another
AGCM is reproducible in modeling science. The CAM5.3, which has been widely
used for the long-term simulation of the climate system, could not efficiently simulate
the eastward propagation of the MJO; instead, the model simulated a tendency for the
MJO to move westward in the IO (Boyle et al., 2015, Jiang et al, 2015). By contrast,
the updated CESM2 with the new CAM6 could realistically simulate the MJO (Ahn et
al., 2020; Danabasoglu et al., 2020). Thus, the well-explored CAM5, which does not
produce a realistic MJO, appears to be a favorable choice for exploring coupling a
simple one-dimensional (1-D) ocean model over the tropical oceans, such as the SIT
model, can improve MJO simulation, as well as the effects of model configuration on
the degree of the improvement. Such a study can also enhance our understanding
regarding the effect of air–sea coupling on the MJO.

The MJO is a tropical atmosphere system that exhibits a more substantial

eastward propagation in boreal winter than in other seasons was the targeted feature in
this study. To examine the sensitivity of MJO simulations to different configurations
of the tropical air–sea coupling, we conducted a series of 30-year numerical
experiments by considering various model configurations (e.g., coupled versus
uncoupled, vertical resolution and depth of the SIT model, coupling domains, and
absence of the diurnal cycle) to investigate the effect of air–sea coupling. This paper
is organized as follows. Section 2 describes the data for validation, the model used for
simulation, and the design of numerical experiments. Section 3 describes the effect of
various tropical air–sea coupling configurations on the MJO simulation determined
through detailed MJO diagnostics. Discussion and conclusions are provided in
Section 4.

**2. Data, methodology, model description, and experimental designs**
**2.1 Data and methodology**

The data analyzed in this study include precipitation from the Global

Precipitation Climatology Project (GPCP), outgoing longwave radiation (OLR) and
daily SST (Optimum Interpolation SST; OISST) from the National Oceanic and
Atmosphere Administration (NOAA), and parameters from the ERA-Interim (ERA-I)
reanalysis (Adler et al., 2003; Dee et al., 2011; Lee et al., 2011; Reynolds and Smith,
1995; Schreck et al., 2018). The SST data for the SIT model were obtained from the
Hadley Centre Sea Ice and Sea Surface Temperature dataset (Rayner et al., 2003;
HadISST1) and the ocean subsurface data (40-layer climatological ocean temperature,
salinity, and currents) for nudging were retrieved from the National Centers for
Environmental Prediction (NCEP) Global Ocean Data Assimilation System (GODAS;
Behringer and Xue, 2004).

We used the CLIVAR MJO Working Group diagnostics package (CLIVAR,

2009) and a 20–100-day filter (Kaylor, 1977; Wang et al., 2014) to determine
intraseasonal variability. MJO phases were defined following the index (namely,
RMM1 and RMM2) proposed by Wheeler and Hendon (2004), which considered the
first two principal components of the combined near-equatorial OLR and zonal winds
at 850 and 200 hPa. The band-passed filtered data were used for calculating the index
and defining phases.

**2.2 Model description**
**2.2.1 CAM5.3**

The CAM5.3 used in this study has a horizontal resolution of 1.9° latitude ×

2.5° longitude and 30 vertical levels with the model top at 0.1 hPa. The MJO could
not be realistically simulated in the CAM5.3. Boyle et al. (2015) demonstrated that
although making the deep convection dependent on SST improved the simulation of
the MJO variance, it exerted a significant negative effect on the mean-state climate of
low-level cloud and absorbed shortwave radiation. By comparing the simulation
results of an uncoupled and coupled CAM5.3, Li et al. (2016) suggested that air–sea
coupling and the convection scheme most significantly affected the MJO simulation
in the climate model.

**2.2.2 1-D high-resolution TKE ocean model**

The 1-D high-resolution turbulence kinetic energy (TKE) ocean model SIT was

used to simulate the diurnal fluctuation of SST and surface energy fluxes (Lan et al.,
2010; Tseng et al., 2014; Tu and Tsuang, 2005). A description of the 1-D high-
resolution ocean model SIT can be found in the appendix. The model was well
verified against in situ measurements on board the R/V Oceanographic Research
Vessel 1 and 3 over the South China Sea (Lan et al., 2010) and on R/V Vickers over
the tropical WP (Tu and Tsuang, 2005).

The SIT model determines the vertical profiles of the temperature and

momentum of a water column from the surface down to the seabed, except in the
fixed ocean model bottom experiment. The default setting of vertical discretization
(e.g., in the control coupled experiment) is 41 layers with 12 layers in the first 10.5 m,
6 layers between 10.5 m and 107.8 m (appendix Fig. A1). In the 1-D TKE ocean
model, temperature and salinity below 107.8 m, where vertical turbulent mixing is
greatly weakened, are nudged toward the climatological values of GODAS data until
4607 m. The extra high vertical resolution is needed to catch detailed temporal
variation of upper ocean temperature characterized by the warm layer and cool skin
(Tu and Tsuang, 2005). To account for the neglected horizontal advection heat flux,
the ocean is weakly nudged (by using a 30-day time scale) between 10.5 m and 107.8
m and strongly nudged (by using a 1-day time scale) below 107.8 m according to the
NCEP GODAS climatological ocean temperature. No nudging is performed within
the upper-most 10.5 m. The SIT model calculates twice for each CAM5 time step (30
min; i.e., coupling 48 times per day).

**2.3 Experimental design**

A series of 30-year numerical experiments (Table 1) were conducted to

investigate the effect of the air–sea interaction on the MJO simulation. The HadSST1
used to force the coupled and uncoupled model was the climatological monthly-mean
SST averaged over 1982-2001. The monthly SST was linearly interpolated to daily
SST fluctuation that forced the model. The SST in the air–sea coupling tropical
region was recalculated by the SIT during the simulation, while the prescribed annual
cycle of SST was used in the areas outside the coupling region. Ocean bathymetry of
the SIT was derived from the NOAA ETOPO1 data (Amante and Eakins, 2009) and
interpolated into $1.9° \times 2.5°$ horizontal resolution.

All simulations were driven by the prescribed annual cycle of SST repeatedly for

30 years. The strategy is to evaluate the simulation capacity of climate models under
the same condition without considering interannual variation induced by SST. This
approach has been widely adopted in many studies (Delworth et al., 2006; Haertel et
al., 2020; Subramanian et al., 2011; Tseng et al., 2014; Wang et al., 2005).

Atmospheric initial conditions and external forcing such as $CO_2$, ozone, and

aerosol in near-equilibrium climate state around the year 2000 were taken from
F_2000_CAM5 component set based on CESM1.2.2 framework development. The
data has been commonly used in present-day simulations using CAM5 (e.g., He et al.,

2017).

The setup of five sets of experiment conducted in this study are described as

follows.
(1) A standalone CAM5.3 simulation forced by climatological monthly HadISST1

(A–CTL) and the control experiment of coupled CAM5–SIT simulation (C–

30NS; 41 vertical levels, coupling in the entire tropics between 30°N and 30°S

with a diurnal cycle). The reasons for tropical coupling are two folds.

Considering that the MJO is essentially a tropical phenomenon, the coupling was

implemented only between 30°N and 30°S. Secondly, coupling a one-

dimensional ocean model in the extratropics without surface flux correction as in

our case would ignore the impacts of strong ocean currents (such as the Kuroshio
and Gulf Stream) and result in large biases.
(2) Upper-ocean vertical resolution experiment: Two simulations with the first layer
centering at 12 m (C–LR12m) and 34 m (C–LR34m). Further details of the
experimental design are shown in appendix Fig. A1. This experiment is to
demonstrate the significant improvement that a fine vertical resolution can
achieve compared to the coarse resolution (e.g., tens of meters) that is often
adopted in slab ocean model.
(3) Shallow ocean bottom experiment: Three simulations with the ocean model
bottom at 10 m (C–HR1mB10m), 30 m (C–HR1mB30m), and 60 m (C–
HR1mB60m) (appendix Fig. A2). Note that all experiments retained the same
vertical resolution (e.g., 1 meter in the first top 10 meters of the ocean) but with
various ocean bottom (i.e., 10, 30, and 60 meters). The purpose is to demonstrate
how the total ocean heat content, which depends on the total depth of the ocean,
can affect the MJO.
(4) Regional coupling experiment: Four simulations with the coupling region in 0°N–
30°N (C–0_30N) and 0°S–30°S (C–0_30S) for latitudinal effect, and 30°E–
180°E (C–30_180E) and 30°E–75°W (C–30E_75W) for longitudinal effect. The
coupling domains are shown in Fig. 1. In this experiment we identified the key
ocean basins where coupling is essential.
(5) Diurnal cycle experiment: To explore the effect of diurnal coupling cycle a non-
diurnal simulation was conducted for a comparison with the C–30NS simulation.
The non-diurnal simulation (C–30NS–nD) considers the air–sea interaction only
once a day, namely, calculating SHF and LHF based on daily mean atmospheric
variables and SST. To prevent the inconsistent local time in different regions, the
coupling frequency at each grid point remained 48 times per day using the same

daily means of atmospheric variables and SST at that particular point. In

contrast, the control simulation calculates air–sea fluxes 48 times a day based on

instantons values. A comparison between the non-diurnal simulation and the

control simulation reveals the effect of diurnal cycle in air–sea coupling.


**3. Results and Discussion**

The realistic simulation of the MJO has always been a major bottleneck in the

development of climate models. In this section, we demonstrate the sensitivity of air–
sea coupling experiments using a 1-D high-resolution ocean model significantly
improves the MJO simulation by the CAM5.3. The period between November and
April when the MJO is the most prominent was the targeted season in this study.

**3.1 Improvement of MJO simulation through air–sea coupling**

This subsection compares the MJO simulation of the control coupled

experiment (C–30NS) with that of the uncoupled AGCM (A–CTL) forced by
climatological monthly SST of HadISST1 to demonstrate the effect of air–sea
coupling on the MJO simulation by coupling the SIT model to the CAM5.3 in the
tropical belt (30°N–30°S).

**3.1.1 Wavenumber–frequency spectra and eastward propagation characteristics**

A wavenumber–frequency spectrum (W–FS) analysis was conducted to quantify

propagation characteristics simulated in different experiments. The spectra
of unfiltered U850 in ERA-I reanalysis, C–30NS, and A–CTL are shown in Fig. 2a–c,
respectively. The C–30NS considering the coupling in 30ºN–30ºS realistically
simulates eastward-propagating signals at zonal wavenumber 1 and 30–80-day
periods (Fig. 2a–b), although with a slightly larger amplitude compared with ERA-I.
By contrast, the uncoupled A–CTL does not yield realistic simulation; instead, it
simulates both eastward (wavenumber 1)- and westward (wavenumber 2)-propagating
signals with an unrealistic spectral shift to time scales longer than 30–80-day.

The major features of the simulated MJO propagation were examined. Figure

2d–f show the time evolution of precipitation and U850 anomalies in Hovmöller
diagrams, which represent lagged correlation coefficients between the precipitation
averaged over 10°S–5°N, 75–100°E and the precipitation and U850 averaged over
10°N–10°S on intraseasonal timescales. Figure 2d indicates eastward propagation for
both precipitation and U850 from the eastern IO to the dateline, with precipitation
leading U850 by approximately a quarter of a cycle. The Hovmöller diagram derived
from the C–30NS (Fig. 2e) exhibits the key characteristics of eastward propagation
for both precipitation and U850 and the relative phases between the two, although the
simulated correlation is slightly weaker than that derived from GPCP and ERA-I. By
contrast, the uncoupled A–CTL simulates intraseasonal signals that propagate
westward over the IO and weak and much slower eastward propagation crossing the
MC and WP (Fig. 2f). The contrast between Fig. 2e and 2f demonstrate that coupling
a 1-D TKE ocean model alone could lead to a significant improvement in an AGCM
in simulating the major characteristics (e.g., amplitude, propagation direction and
speed, and phase relationship between precipitation and circulation) of the MJO.

**3.1.2 Coherence of the simulated MJO**

Cross-spectral analysis was conducted to examine the coherence and phase lag

between tropical circulation and convection, which were plotted over the tropical
wave spectra. Figure 2g–i show the symmetric part (e.g., Wheeler and Kiladis, 1999)
of OLR and U850 in ERA-I/NOAA data, C–30NS, and A–CTL, respectively. We
present only the spectra between 0 to 0.35 day$^{-1}$ to highlight the MJO and equatorial
Kelvin waves. The most prominent characteristics seen in ERA-I/NOAA data are the
peak coherence at wavenumbers 1–3 and a phase lag of approximately 90° in the 30–
80-day band (Ren et al., 2019; Wheeler and Kiladis 1999). The coupled experiment
C–30NS simulates strong coherence in this low-frequency band (wavenumber 1) and
exhibits a realistic phase lag relationship between U850 and OLR perturbations.
However, the coherence at wavenumbers 2–3 for the 30–80-day period simulated by
C–30NS is weaker than that in ERA-I/NOAA data. This undersimulation was also
noted in CCSM4 (Subramanian et al., 2011), the uncoupled and coupled CAM4 and
CAM5 (Li et al., 2016), and NorESM1-M (Bentsen et al., 2013), which had a version
of the CAM as an AGCM. In summary, C–30NS considering the coupling between
30ºN–30ºS produces coherent and energetic patterns in the eastward-propagating
intraseasonal fluctuations of U850 and OLR in the tropical IO and WP that are
generally consistent with the MJO characteristics. By contrast, the MJO
characteristics in A–CTL are considerably weaker than those in C–30NS and that in
ERA-I/NOAA data.

### 3.1.3 Horizontal and vertical structures of the MJO across the MC

Figure 2j–o show the horizontal and vertical structures of the MJO when deep
convection is the strongest over the MC (i.e., phase 5). Figure 2j–l present the 20–
100-day filtered OLR (W m$^{-2}$, shaded) and 850-hPa wind (m s$^{-1}$, vector). C–30NS
realistically simulated the enhanced tropical convection over the eastern IO and the
Kelvin-wave-like easterly anomalies over the tropical WP despite undersimulating
the convection over the MC (Fig. 2j and 2k). By contrast, A–CTL failed to simulate
the enhanced convection over the eastern IO and MC; instead, it simulated
considerably weaker convection and easterly winds over the MC and WP,
respectively, than that in ERA-I/NOAA data (Fig. 2j and 2l).
Figure 2m–o show the vertical–longitudinal profiles of 20–100-day filtered
15°N–15°S averaged vertical velocity (OMEGA; Pa s$^{-1}$, shaded) and moist static
energy tendency (dMSE/dt) anomalies (W m$^{-2}$, contour) at phase 5. The spatial
distribution of negative OMEGA (ascending motion) anomalies generally agreed with
OLR anomalies in C–30NS simulation and NOAA data over the Indo-Pacific region
(Fig. 2m and 2n). The relatively spatial relationship between the ascending motion
and dMSE/dt seen in ERA-I is well simulated in the coupled experiment C–30NS. For
example, positive dMSE/dt anomalies on the eastern side of the anomalous ascent
demonstrate that the energy recharge process occurs in advance of the MJO
convection over the lower-tropospheric easterlies (Fig. 2m and 2n), whereas negative
dMSE/dt anomalies on the western side reveal that the discharge process occurs
during and after convection over the lower-tropospheric westerlies. By contrast, this
phase relationship, considered to be an essential feature leading to the eastward
propagation of an MJO (Hannah and Maloney 2014; Heath et al., 2021), is not
properly simulated in the uncoupled experiment A–CTL (Fig. 2o), in which the
simulated weak negative OMEGA is located between negative and positive dMSE/dt
anomalies over weak lower-tropospheric wind anomalies and associated with weak
convection over the MC (Fig. 2l).
The temporal evolution of NOAA OLR and ERA-I U850 (Fig. 3a) indicates
that convection originating in the western IO is enhanced during its eastward
propagation to the MC where it reaches the peak amplitude and then gradually
weakened when continuing moving eastward to the dateline. In the coupled
experiment C–30NS, this evolution of convectively coupled circulation is realistically
simulated, although it is weaker than the strength seen in NOAA OLR (Fig. 3b).
Moreover, the split of convection into two cells off the equator in phase 6 is
appropriately simulated in C–30NS (P6 in Fig. 3a and 3b). This split was caused by
the topographic and land–sea contrast effects of the MC (Tseng et al., 2017).
Associated with the split is the southward detouring of the anomalous convection
during the passage of the MJO through the MC (Kim et al. 2017, Tseng et al., 2017;
Wu and Hsu, 2009). After the passage of the MJO through the MC, the anomalous
convection stays south of the equator and continues moving eastward to the
dateline. In the uncoupled A–CTL, the systematic eastward propagation of
convectively coupled MJO circulation from the IO into the MC is not simulated.
Instead, the convection over the MC develops in situ at a later stage than that
observed (e.g., P6 in Fig. 3c) and dissipated rapidly. The A–CTL simulates a pair of
off-equator convection anomalies in the eastern IO during phase 2 (P2 in Fig. 3c) that
moves westward toward the central IO and were amplified at later stages (e.g., P4 in
Fig. 3c). This unrealistic evolution explains the westward propagation tendency
observed in the Hovmöller diagram (Fig. 2f).

**3.1.4 Characteristics of air–sea interaction**

Figure 4a–c show the longitude–phase diagram in which the 20–100-day filtered

precipitation (shaded) and SST (contour) anomalies were averaged over 10°S–10°N to
determine the relationship between precipitation and SST fluctuations and to establish
a link between air–sea coupling and convection. The propagation of the enhanced
convection with positive SST anomalies to the east could be clearly seen in
GPCP/OISST and the coupled experiment C–30NS (Fig. 4a and 4b). The highest SST
anomaly (SSTA) leads the maximum precipitation anomaly by approximately 2–3
phases, and the SSTA begins to decrease following the onset of enhanced
precipitation. The ERA-I and OISST data reveal the following relationship between
net surface flux and SST: the decreased (increased) LHF/SHF and increased
(decreased) downward radiation flux leading (lagging) the positive (negative) SSTA
east (west) of anomalous deep convection. This well-known lead–lag relationship
reflecting the active air–sea interaction in an MJO is realistically simulated in the
coupled experiment C–30NS (not shown).

The contrast between C–30NS and A–CTL confirms the key role of the air–sea

interaction in contributing to the eastward propagation and demonstrates that the
eastward propagation simulation can be markedly improved by incorporating the air–
sea interaction process in the model, even when using a simple 1-D ocean model such
as SIT.

### 379     3.1.5 Vertically tilting structure

The warm SST was the key forcing that contributed to the boundary layer

convergence before the onset of deep convection (Li et al., 2020; Tseng et al., 2014).
Hence, the warmer upper ocean enhances the low-level atmospheric convergence and
then leads to enhanced low-level moisture and preconditioned deep convection and
eastward propagation. This moistening process associated with warm ocean surface
temperature is well simulated in the coupled experiment C–30NS but is not shown
here. Instead, we present the coupling of moisture divergence (MD) and atmospheric
circulation.

MD and zonal wind anomalies from the surface to the upper troposphere

averaged over the 10°S–10°N and 120–150°E region are shown in Fig. 4d–f to depict
the relationship between the vertically tilting structure of MD and zonal wind
anomalies. Note that the active convection occurred around phase 5. The coupled
experiment C–30NS (Fig. 4e) realistically simulates the deepening of coupled MD
and zonal wind anomalies with time (Fig. 4d). An evolution from the right to left
seen in each panel of Fig. 4d–f is equivalent to the eastward movement of vertically
tilting circulation from the eastern IO into the MC because of the eastward-
propagating nature of the MJO. Figure 4d and 4e show that in both ERA-I reanalysis
and the coupled experiment C–30NS, the near-surface convergence (negative MD)
occurring in the easterly anomalies lead the convection and continued deepening up
to 500 hPa from phase 2 to phase 6 when the easterly anomalies switch to westerly
anomalies. By contrast, this evolution of coupled MD–zonal wind anomalies are not
appropriately simulated in the uncoupled experiment (Fig. 4f). For example, a slow
deepening with time is observed in the MD anomaly but not in the zonal wind
anomaly that exhibits a vertically decayed structure, suggesting that MD and wind
anomalies are not well coupled, as noted in the ERA-I/NOAA data and the control
coupled experiment.

In the ERA-I reanalysis data, the negative near-surface MD anomalies appear

first under the easterly anomaly and continue deepening between the easterly and
westerly anomalies. This development in the phase relationship between MD and
zonal wind anomalies in both ERA-I reanalysis data and the coupled simulation is
consistent with the well-known structure embedded in the MJO, namely the near-
surface convergence in the easterly phase (i.e., a boundary-layer moistening process;
Kiranmayi and Maloney 2011; Li et al., 2020; Tseng et al., 2014), followed by the
deep convection when transitioning to the westerly phase. This close phase
relationship that is key to the eastward propagation is appropriately simulated in the
coupled experiment but not in the uncoupled experiment.

### 417     3.1.6 Intraseasonal variance of precipitation

Figure 4g–i present the spatial distribution of intraseasonal variance of

precipitation. In the GPCP data, the maximum variance is noted over the tropical
eastern IO, MC, and tropical WP. The maximum variance south of the island in the
MC and the equator in the tropical WP reflects the southward shift of the MJO deep
convection when passing through the MC, partly due to the blocking effect of
mountainous islands and the higher moisture content over high SST south of the
equator in the region during boreal winter (Kim et al., 2017; Ling et al., 2019; Sobel
et al., 2008; Tseng et al., 2017; Wu and Hsu, 2009). Although the control coupled
experiment fails to simulate the variance maximum in the tropical eastern IO, it
appropriately simulates the maximum variance over the tropical WP, reflecting its
ability to simulate the eastward propagation of the MJO through the MC. By contrast,
the uncoupled A–CTL experiment simulates considerably weaker intraseasonal
variance in both the tropical eastern IO and the tropical WP. Figure 4j–l are the 20–
100-day filtered SST (K, shaded) and 850-hPa wind (m s$^{-1}$, vector) during MJO
phase 7 when deep convection is the strongest over the dateline. The coupled
experiment C–30NS realistically simulates the negative SST anomaly over the MC
and WP when enhanced tropical convection passed through the MC to the dateline,
indicating the capability of the SIT model to reproduce the SST anomaly by
exchanging LHF/SHF between the atmosphere and ocean. In A-CTL, no SST
anomaly is evident because the model was forced by prescribed climatological SST.
The contrast seen in Fig. 4j–l demonstrates the essential role of atmosphere–ocean
coupling in shaping the MJO. A delayed air–sea interaction mechanism was noted,
where a preceding active-phase MJO may trigger an inactive-phase MJO through the
delayed effect of the induced SST anomaly. In addition, the westerly winds at 850
hPa moving southward between MC and WP are captured by the control experiment
C–30NS and are similar to the ERA-I reanalysis winds (Fig. 4j and 4k). By contrast,
A–CTL forced by climatological monthly SST (<0.05 K phase$^{-1}$ anomaly) fails to
simulate the southward westerly wind of the region extending from the MC to the
dateline (Fig. 4l).

## 3.2 Effect of upper-ocean vertical resolution

In the control coupled experiment C–30NS, the vertical resolution in the upper 10.5 m was 1 m. Tseng et al. (2014) suggested that fine vertical resolution is crucial for appropriately simulating the eastward propagation. To investigate the effect of vertical resolution, two experiments with a thicker first layer were conducted by moving the center of the layer to 11.5 m (C–LR12m) and 33.9 m (C–LR34m), respectively, as opposed to the control experiment in which 10 layers were implemented in the first 10.5 meters (see appendix Fig. A1 for vertical discretization). The dramatic changes in vertical profile of ocean temperature between the fine and coarse resolution simulation are demonstrated in Fig.5, which presents the 20–100-day filtered oceanic temperature anomalies (K, shaded) between 0 and 60 m depth for MJO phase 1, 3, 5, and 7. The amplitude of ocean temperature is the largest in C–30NS and much weaker in C–LR12m and C–LR34m. In addition, there is a clear vertical stratification of ocean temperature in C–30NS, whereas C–LR12m and C–LR34m are well mixed because of not vertically gridded. This demonstrates the necessity of fine vertical gridding for resolving the quick fluctuation of ocean temperature when interacting with the atmosphere.

The W–FS spectral peaks of U850 in C–LR12m are concentrated in eastward-propagating wavenumber 1 at three timescales (e.g., longer than 80 days, 30–80 days, and approximately 30 days; Fig. 6a). In C–LR34m, both eastward and westward signals are simulated with the dominant W–FS timescale longer than 80 days (Fig. 6b). The appearance of both eastward and westward signals at a lower frequency implied a stronger stationary tendency or weaker eastward-propagating tendency. This result is consistent with that reported by Tseng et al. (2014) that the scientific reproducibility of coarser resolution causes a longer intraseasonal periodicity and slower eastward propagation of the MJO.

The effect of vertical resolution on the MJO simulation can be seen in the
Hovmöller diagram. The eastward propagation simulated in C–LR12m (Fig. 6c)
markedly weakened after crossing the MC compare with that simulated in the control
experiment C–30NS (Fig. 2e). In C–LR34m, the quasi-stationary fluctuation and
westward propagation are simulated over the IO (Fig. 6d), appearing similar to those
in A–CTL (Fig. 2f). The lead–lag relationship between precipitation (zonal wind) and
SST is poorly simulated in C–LR12m (Fig. 6e) and even more poorly simulated in C–
LR34m (Fig. 6f). This result confirms the finding reported by Tseng et al. (2014) that
a higher vertical resolution in the upper few meters below the surface allows for a
faster air–sea interaction, thus resulting in a more realistic simulation of the MJO.

**3.3 Effect of the lowest boundary of the SIT model**
The ocean is a vital energy source for the MJO. Although vertical resolution is
crucial for the efficiency of air–sea interaction, the thickness of the upper ocean that
interacts with the atmosphere represents the ocean heat content to substantiate the
MJO. A key question is how the total ocean heat content, which depends on the total
depth of the ocean, can affect the MJO. Considering two models with the same
vertical resolution, the model with thinner ocean (e.g., 10 meter) would interact as
efficiently as another model with thicker ocean (e.g., 60m) but with much less heat to
release to or to absorb from the atmosphere. The former would have less impact on
the atmosphere than the latter. Using the same vertical resolution, three experiments
with various ocean depths ocean bottom at 10, 30, and 60 m were conducted (see
appendix Fig. A2 and Table 1).
The spectra and the Hovmöller diagrams shown in Fig. 7a–c and Fig. 7d–f,
respectively, demonstrate that the thicker ocean model simulates a stronger MJO with
a frequency closer to those in the coupled experiment C–30NS and ERA-I/NOAA
data, and more realistic eastward propagation. In addition, the lead–lag relationship
between precipitation (wind) and SST is more realistically simulated with increasing
thickness of the ocean model (Fig. 7g–i).
This result suggests that the thickness of the vertically-gridded ocean that interacts
with the atmosphere strongly affects the frequency of the simulated MJO. A thinner
(thicker) vertically-gridded ocean is more quickly (slowly) recharged and discharged
through SHF and LHF exchange between the atmosphere and ocean and therefore
likely fluctuates at a faster (slower) tempo. The simulated periodicity is therefore
affected by the thickness (or ocean heat content) of upper ocean that interacts
rigorously with the atmosphere. Although the result suggests 60 m is an appropriate
thickness to realistically simulate the periodicity of the MJO, we did not intend to
suggest the exact thickness required for a proper simulation because it might depend
on the model. The upper ocean should be adequately thick to contain a certain amount
of heat to generate appropriate periodicity. However, the reason for the intraseasonal
time scale (i.e., 20-100 days) should be determined in future studies. This finding
does not suggest a constant periodicity because periodicity might be affected by the
time-varying structure of the atmosphere and ocean in the real world.

**3.4 Effects of coupling domains**
The MJO is a planetary-scale phenomenon. Given its large-scale circulation, the
air–sea interaction affecting the MJO likely occurs in a much larger area than the
region near the major convection anomalies. In this section, we discuss the effect of
coupling domain on model ability to simulate the eastward propagation speed and
periodicity of the MJO. Four experiments considering the coupling in various
domains (C–0_30N, C–0_30S, C–30_180E, and C–30E_75W, Fig. 1) were conducted
for the purpose. The results are shown in Fig. 8. The C–0_30N that considered the
coupling in the tropics between the equator and 30°N simulates the least realistic MJO
propagation in terms of W–FS (Fig. 8a), zonal wind–precipitation coupling (Fig. 8e),
and SST–precipitation (Fig. 8i) among the four regional coupling experiments. By
contrast, coupling only the tropics between the equator and 30°S simulates a more
realistic MJO in all three aspects (i.e., spectrum in Fig. 8b, temporal evolution of
precipitation/wind, and precipitation/SST coupling in Fig. 8f and 8j). Figure. 9a
indicates that the negative OLR anomalies at phase 5 simulated in C–0_30N stays
mainly north of the equator and does not shift southward in the MC as revealed in
ERA-I reanalysis and NOAA OLR and in the control experiment C–30NS, and the
convection over the IO is unrealistically weak. By contrast, the southward detouring
in the MC is realistically simulated in C–0_30S that coupled only the tropical ocean
between the equator and 30°S. This result indicates that air–sea coupling occurring
south of the equator is the key to producing appropriate eastward propagation and
detouring of the MJO through the MC. Without this coupling, the C–0_30N
experiment fails to realistically simulate the eastward propagation of the MJO (Fig.
8e). This contrast can be attributed to the warmer ocean surface and higher moisture
content found south of the equator in boreal winter, which comprise a more favorable
environmental condition for air–sea coupling and convection–circulation coupling and
the occurrence of the MJO.

MJO simulations can be affected by air–sea coupling in the longitudinal domain.

Tseng et al. (2014) examined this effect by allowing coupling in different regions
(e.g., the IO, WP, and IO + WP) and found that the IO + WP coupling experiment
yielded the most satisfactory MJO simulation in terms of the zonal W–FS and
eastward propagation characteristics. In this study, we conducted sensitivity
experiments in which we allowed coupling in the tropics in two longitudinal domains,
namely 30°E–180°E (C–30_180E) and 30°E–75°W (C–30E_75W). The 30°E–180°E
region covered the IO and WP, and the 30°E–75°W region covered the IO and the
entire tropical Pacific. As shown in Fig. 8, the C–30E_75W experiment simulates
more realistic MJO than the C–30_180E experiment, with stronger eastward
propagation and larger amplitudes in the spectrum (Fig. 8c and 8d) and Hovmöller
diagrams of precipitation/wind (Fig. 8g and 8h) and precipitation/SST (Fig. 8k and
8l). The simulated MJO in C–30E_75W propagated farther east than that in C–
30_180E, particularly evident in Fig. 8k and 8l. The spatial distributions of circulation
and OLR shown in Fig. 9c and 9d indicate the presence of a stronger convective-
coupled circulation system over the MC and WP in C–30E_75W. These results
suggest that coupling over the entire tropical IO and Pacific could enhance the
strength and eastward propagation of the MJO and encourage farther propagation to
the central Pacific.

**3.5 Diurnal versus no diurnal cycle in air–sea coupling**

Previous studies showed that the diurnal cycle in the MC can weaken the MJO

and its eastward propagation (Hagos et al., 2016; Oh et al., 2013). We conducted an
experiment to determine whether computing surface heat fluxes using daily mean
values, instead of instantaneous values, of atmospheric variables and SST with the
same coupling frequency would affect the MJO simulation. The coupling in the model
was conducted through the SHF and LHF exchange between the atmosphere and
ocean, that were calculated based on simulated winds, moisture, and temperature. As
mentioned in Section 2.3, air–sea fluxes were calculated twice for every time step
(coupling 48 times per day) in the control coupled experiment (C–30NS) based on the
instantaneous values of atmospheric and oceanic variables. In the experiment in which
the diurnal cycle was removed (C–30NS–nD), air–sea fluxes were calculated as in C–
30NS but were based on daily means of both atmospheric variables and SST. Doing
this removed certain diurnal effects of air-sea coupling. The results shown in Fig. 10
reveal the enhancement of the eastward-propagating signals in the MJO (e.g., a larger
amplitude in spectrum; Fig. 10a) and further eastward propagation (Fig. 10b) as well
stronger coupling between precipitation and SST (Fig. 10c) in C–30NS–nD. The
overall results are consistent with previous finding that the diurnal cycle tends to
reduce the amplitude of the MJO, indicating that the weakening effect occurs through
air–sea coupling in addition to those processes in the atmosphere. Previous studies
have hypothesized that rapid interaction processes in the diurnal time scale tend to
extract energy from the MJO, thus reducing the strength and propagation tendency of
the MJO. However, a comparison between the spectra of C–30NS and C–30NS–nD
indicates that the experiment in which the diurnal cycle is removed appeared to
oversimulate the MJO with unrealistic strength, suggesting that the effect of the
diurnal cycle should be considered in the model to simulate a more realistic MJO.
However, whether this is a common result in different models remain to be examined.

**4. Discussion and conclusions**

Air–sea coupling is a key mechanism for the successful simulation of the MJO

(Chang et al., 2019; DeMott et al., 2015; Jiang et al., 2015, 2020; Kim et al., 2010; Li
et al., 2016; Li et al., 2020; Newman et al., 2009; Tseng et al., 2014). This study,
following the study of Tseng et al. (2014), demonstrated that coupling a high-
resolution 1-D TKE ocean model (namely the SIT model) to the CAM5, namely the
CAM5–SIT, significantly improved the MJO simulation over the standalone CAM5.
By coupling SIT model to an AGCM different from Tseng et al. (2014), this study
confirms the scientific reproducibility for the improvement of MJO simulation in
modeling science. The CAM5–SIT realistically simulates the MJO characteristics in
many aspects (e.g., intraseasonal periodicity, eastward propagation, coherence in the
low-frequency band, detouring propagation across the MC, tilting vertical structure,
and intraseasonal variance in the WP).

Systematic sensitivity experiments were conducted to investigate the effects of

the vertical resolution and the thickness of the 1-D ocean model, coupling domains,
and the absence of the diurnal cycle. The results of all the sensitivity experiments are
summarized in Fig. 11a and 11b, which show four common metrics for MJO
evaluation. The four metrics are the propagation speed of the MJO (estimated from
the U850 Hovmöller diagram as Fig. 2d–f) versus the power ratio of eastward- and
westward-propagating 30–80-day signals (E/W ratio, derived from the zonal W–FS)
in Fig. 11a and the eastward propagation speed of the 30–80-day filtered precipitation
anomaly (estimated from the precipitation Hovmöller diagram) versus the variance
explained by RMM1 and RMM2 (i.e., the sum of the variance explained by the first
two empirical orthogonal functions (EOF1 and EOF2) based on Wheeler and Hendon,
2004) in Fig. 11b. Based on the maximum precipitation anomaly and zero values of
U850 (indicating deep convection region), propagation speeds of precipitation and
U850 were calculated from Hovmöller diagrams between 60°E and 150°W. Overall,
the control experiment C–30NS simulates the most realistic MJO among all
sensitivity experiments.

As for vertical resolution, we determined that the MJO simulation efficiency

decreased when the vertical resolution of the SIT model is decreased from 1 m to 11.5
or 33.9 m, as simulated in the C–LR12m and C–LR34m experiments, respectively.
This finding, consistent with that reported by Tseng et al. (2014), suggests that a finer
vertical resolution more effectively resolves temperature variations in the ocean warm
layer and enhances atmospheric–ocean coupling, thus enabling the upper ocean to
more efficiently respond to atmospheric forcing by providing sensible and latent heat
fluxes; this results in superior synchronization between the lower atmosphere and the
upper ocean.

We observed that the shallower ocean model bottom could speed up the eastward

propagation of the MJO by producing more perturbations of shorter periodicity (Fig.
7) and results in a weaker MJO. The shallower ocean layer with vertical grids likely
responds more quickly to atmospheric forcing but provides less sensible and latent
heat fluxes to the atmosphere. Thus, the MJO propagates too fast with a weaker
amplitude.

In the coupling domain sensitivity experiments, we investigated the essential

coupling domain required to simulate the realistic MJO and the effect of the domain
on the MJO simulation. Coupling only the northern tropics fails to simulate the
eastward propagation, whereas coupling only the southern tropics yields a more
realistic MJO simulation, although this simulation is inferior to coupling the entire
tropics. This contrast reveals the importance of the southern tropical ocean, especially
in the MC where high SST and moisture content are noted. Coupling in the southern
tropics is therefore essential for providing the energy required to maintain the MJO
and its eastward propagation. By contrast, the northern tropics are relatively dry and
cool. Coupling in this region is therefore less effective in improving MJO simulation.

In the longitudinal domain sensitivity experiments, we found that the MJO

amplitude and the eastward extend of its eastward propagation are enhanced by
extending the eastern boundary of the coupling domain from the tropical eastern IO to
the tropical WP and further to the tropical eastern Pacific (Fig. 1). Further extension
of the domain to cover the tropical Atlantic does not exhibit further enhancement (not
shown). This result indicates that coupling in the tropical central and eastern Pacific,
although not the major MJO signal regions (i.e., from the tropical IO to the tropical
WP), still played a marked role in sustaining the MJO. We propose the following to
explain this effect. Because of the planetary scale of the MJO, the near-surface
easterly circulation to the east of the convection core often extended to the tropical
central and eastern Pacific where the climatological easterly prevailed. The coupling
beyond the WP increased low-level moisture transport and convergence to the east of
the convection and establish an environment suitable for the further eastward
propagation of the MJO. This effect was likely terminated by the landmass of Central
America when the tropical Atlantic was further included. Thus, a further eastward
extension of the coupling domain exerted little effect on further enhancing the MJO. A
diagnostic study on the effect of the longitudinal coupling domain is being conducted,
and the results will be reported in a following paper.
The diurnal versus nondiurnal cycle experiment indicates that nondiurnal
coupling tended to enhance eastward-propagating signals but slow down the eastward
propagation (Fig. 11a–b). This result is consistent with the finding of previous studies
that the diurnal cycle in the atmosphere extracts energy from the MJO, thus
weakening it.
In this study, we demonstrated how air–sea coupling can improve the MJO
simulation in a GCM. The findings are as follows.
(1) Better resolving the fine structure of the upper-ocean temperature and therefore
the air–sea interaction leads to more realistic intraseasonal variability in both
tropical SST and atmospheric circulation.
(2) An adequate thickness of vertically-gridded upper ocean is required to simulate a
delayed response of the upper ocean to atmospheric forcing and lower-frequency
fluctuation.
(3) Coupling the tropical eastern Pacific, in addition to the tropical IO and the tropical
WP, can enhance the MJO and facilitate the further eastward propagation of the
MJO to the dateline.
(4) Coupling the southern tropical ocean, instead of the norther tropical ocean, is
essential for simulating a realistic MJO.
(5) Stronger MJO variability can be obtained without considering the diurnal cycle in
coupling.
Our study confirmed the effectiveness of air–sea coupling for improving MJO
simulation in a climate model and demonstrated how and where to couple. The
findings enhance our understanding of the physical processes that shape the
characteristics of the MJO.

*Code and data availability.* The model code of CAM5–SIT is available at
https://doi.org/10.5281/zenodo.5510795. Input data of CAM5–SIT using the
climatological Hadley Centre Sea Ice and Sea Surface Temperature dataset and
GODAS data forcing, including 30-year numerical experiments, are available at
https://doi.org/10.5281/zenodo.5510795.

*Author contributions.* HHH is the initiator and the primary investigator of the
Taiwan Earth System Model project. YYL is the CAM5–SIT model developer and
writes the majority part of the paper. WLT and LCJ assist in MJO analysis.

*Competing interests*. The authors declare that they have no conflict of interest.

*Acknowledgements.* The contribution from YYL, HHH, WLT, and LCJ to this study is
supported by Ministry of Science and Technology of Taiwan under contracts MOST
110-2123-M-001-003, MOST 110-2811-M-001-603, MOST 109-2811-M-001-624
and MOST108-2811-M-001-643. Our deepest gratitude goes to the editors and
anonymous reviewers for their careful work and thoughtful suggestions that have
helped improve this paper substantially. We sincerely thank the National Center for
Atmospheric Research and their Atmosphere Model Working Group (AMWG) for
release CESM1.2.2. We thank the computational support from National Center for
High530 performance Computing of Taiwan. This manuscript was edited by Wallace
Academic Editing.

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

Table 1. List of experiments

| Section | Category | Experiments | Description |
|---------|----------|-------------|-------------|
| 3.1 | Coupled or uncoupled | A–CTL | Standalone CAM5.3 forced by forced by the monthly mean Hadley Centre SST dataset version 1 climatology |
| | | C–30NS (the control coupled experiment) | CAM5.3 coupled with SIT over the tropical domain (30°N–30°S), with 41 layers of finest vertical resolution (up to the seabed) and diurnal cycle; the frequency of CAM5 being exchanged with CPL is 48 times per day |
| 3.2 | Upper-ocean vertical resolution | C–LR12m | The first ocean vertical level starts at 11.5 m with 31 layers (beside SST and cool skin layer are 11.5 m, 29.5 m and 43.6 m up to the seabed) |
| | | C–LR34m | The first ocean vertical level starts at 33.9 m with 28 layers (beside SST and cool skin layer are 33.9 m, 76.9 m and 96.8 m up to the seabed) |
| 3.3 | Lowest boundary of SIT | C–HR1mB10m | The lowest boundary of SIT has a depth of 10 m (model depth between 0 m and 10 m) |
| | | C–HR1mB30m | The lowest boundary of SIT has a depth of 30 m (model depth between 0 m and 30 m) |
| | | C–HR1mB60m | The lowest boundary of SIT has a depth of 60 m (model depth between 0 m and 60 m) |
| 3.4 | Regional coupling domain in latitude | C–0_30N | Coupled in the tropical northern hemisphere (0°N–30°N, 0°E–360°E) |
| | | C–0_30S | Coupled in the tropical southern hemisphere (0°S–30°S, 0°E–360°E) |
| | Regional coupling domain in longitude | C–30_180E | Coupled in the Indo-Pacific (30°N–30°S, 30°E–180°E) |
| | | C–30E_75W | Coupled over the Indian Ocean and Pacific Ocean (30°N–30°S, 30°E–75°W) |
| 3.5 | Absence of the diurnal cycle | C–30NS–nD | Absence of the diurnal cycle in C–30NS; the CAM5.3 daily atmospheric mean of surface wind, temperature, total precipitation, net surface heat flux, u-stress and v-stress over water trigger the SIT and daily mean SST feedback to atmosphere; the frequency of CAM5 is exchanged with CPL 48 times per day |

Experiment abbreviations: "A" means standalone AGCM simulation. "C" means the
CAM5.3 coupled to the SIT model.

zonal wind (contoured, m s$^{-1}$) averaged over 10°N–10°S, 120–150°E; solid, dashed, and thick-black curves are positive, negative, and zero values, respectively. (g)–(i) Variation of 30–60-day filtered precipitation in the eastern IO and the WP in observation (color shading), and the ratio between intraseasonal and total variance (contoured) and (j)–(l) composites 20–100-day filtered SST (K, shaded) and 850-hPa winds (m s$^{-1}$, vector) at phase 7 when deep convection was the strongest over the dateline. Reference vector shown at the top right corner of each panel. (a), (d), (g), and (j) are from the ERA-I/NOAA data; (b), (e), (h), and (k) are from the control coupled experiment C–30NS; and (c), (f), (i), and (l) are from the uncoupled experiment A–CTL.

**Figure 5.** Composites of 20–100-day filtered oceanic temperature (K, shaded) between 0 and 60 m depth for MJO phase 1, 3, 5, and 7 (shown at the lower right corner of each panel) in C–30NS, C–LR12m and C–LR34m.

**Figure 6.** (a)–(b) Same as in Fig. 2(a) but for the C–LR12m and C–LR34m. (c)–(d) Same as in Fig. 2(d) but for the C–LR12m and C–LR34m. (e)–(f) Same as in Fig. 4(a) but for the C–LR12m and C–LR34m.

**Figure 7.** Same as in Fig. 6 but for the C–HR1mB10m, C–HR1mB30m, and C–HR1mB60m.

**Figure 8.** Same as in Fig. 6 but for the C–0_30N, C–0_30S, C–30_180E, and C–30E_75W.

**Figure 9.** Same as in Fig. 3 but for phase 5 in the C–0_30N, C–0_30S, C–30_180E, and C–30E_75W.

**Figure 10.** Similar as in Fig. 6 but for the C–30NS–nD.

**Figure 11.** Scattered plots of various MJO indices in the ERA-I/NOAA data and 12 experiments: (a) power ratio of east/west propagating waves of wavenumber 1–3 of 850-hPa zonal winds (X-axis) with a 30–80-day period and eastward propagation speed of U850 anomaly (Y-axis) from the Hovmöller diagram and (b) RMM1 and RMM2 variance and eastward propagation speed of the filtered precipitation anomaly derived from the Hovmöller diagram.

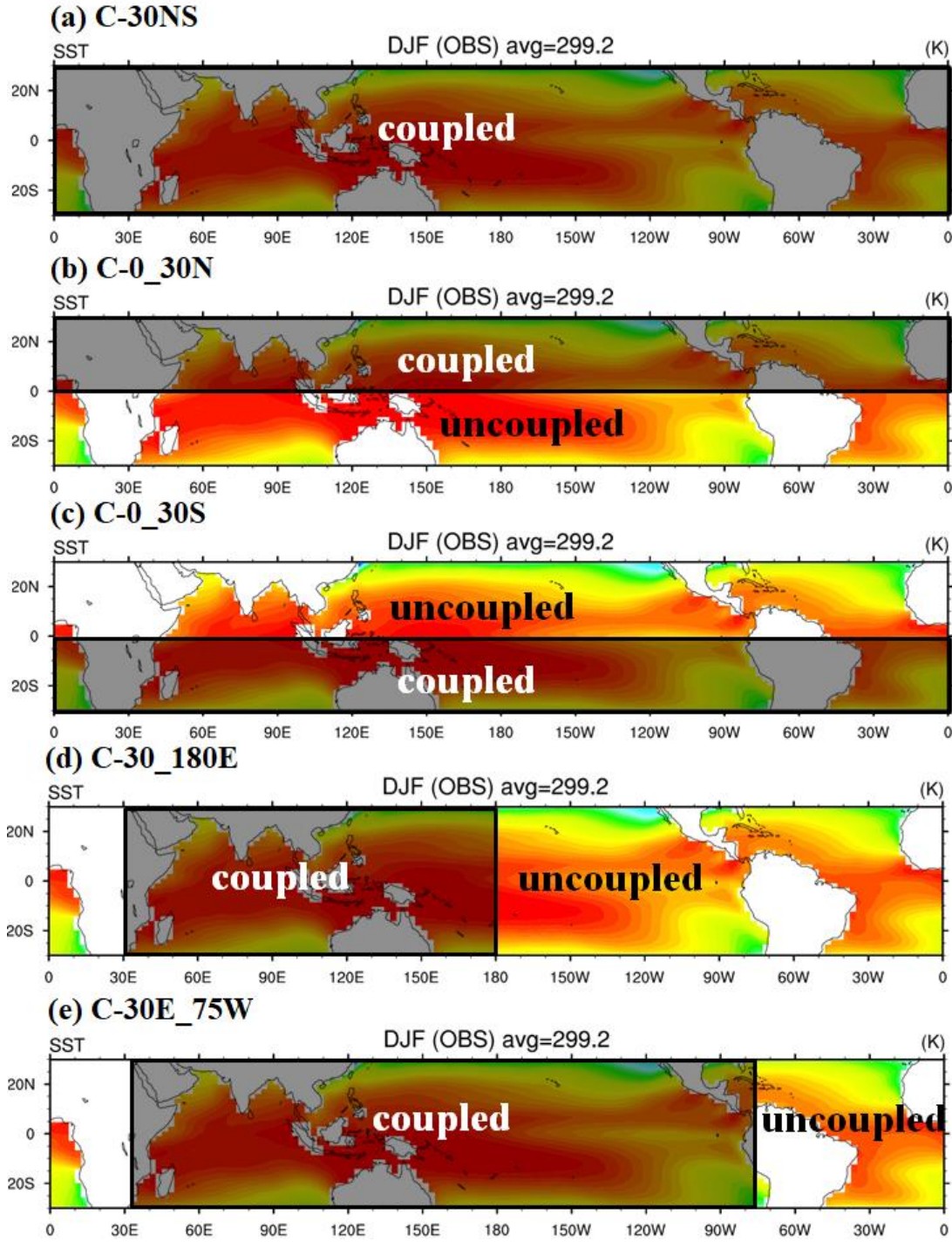

**Figure 1.** Schematics of coupled and uncoupled domains in the regional coupling experiment: (a) C–30NS, (b) C–0_30N, (c) C–0_30S, (d) C–30_180E, and (e) C–30E_75W. The background is the climatological mean SST in December–February (DJF).

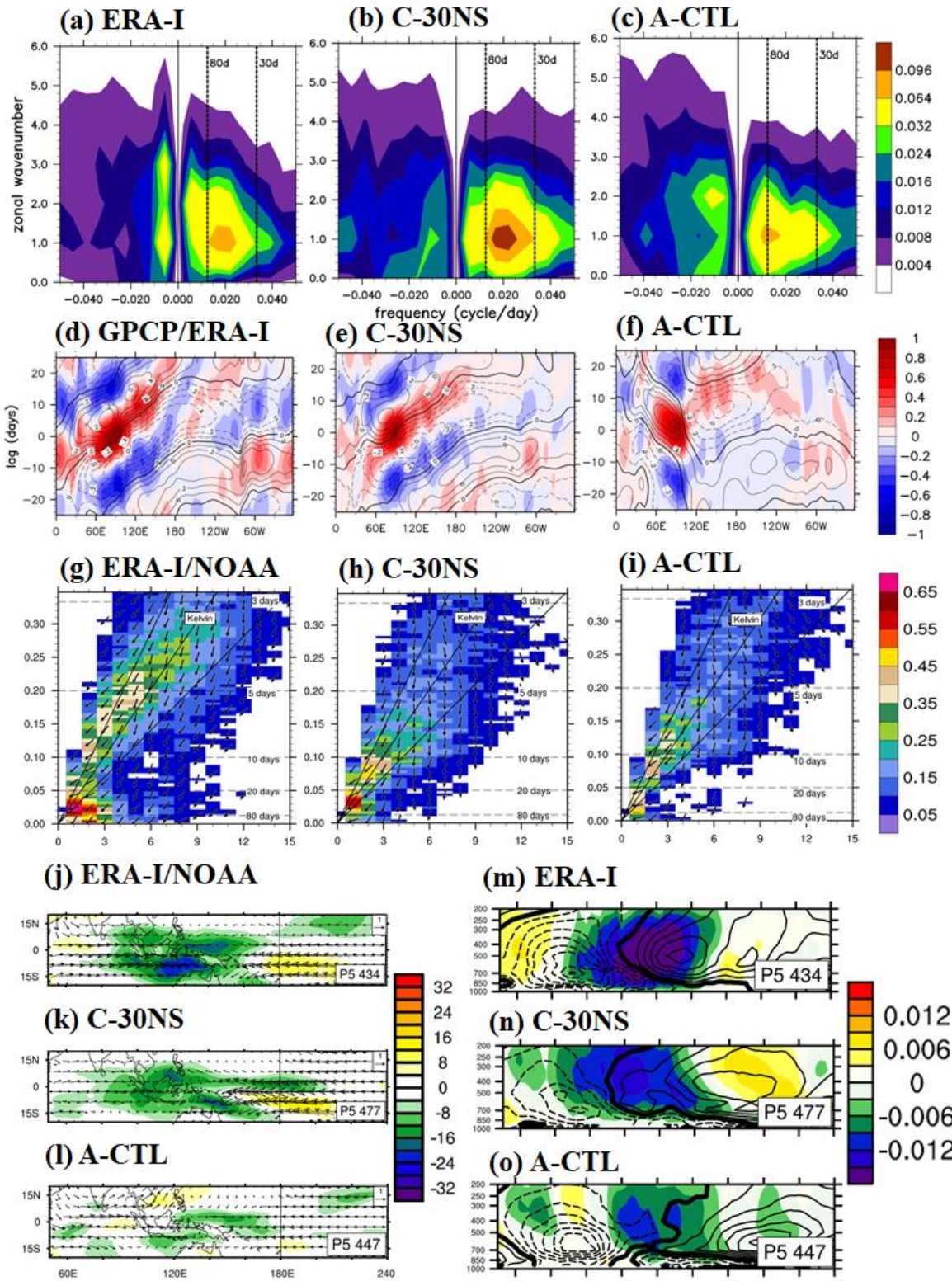

**Figure 2.** (a)–(c) Zonal wavenumber–frequency spectra for 850-hPa zonal wind averaged
over 10°S–10°N in boreal winter after removing the climatological mean seasonal cycle.
Vertical dashed lines represent periods at 80 and 30 days, respectively. (d)–(f) Hovmöller

diagrams of the correlation between the precipitation averaged over 10°S–5°N, 75–100°E and the intraseasonally filtered precipitation (color) and 850-hPa zonal wind (contour) averaged over 10°N–10°S. (g)–(i) Zonal wavenumber–frequency power spectra of anomalous OLR (colors) and phase lag with U850 (vectors) for the symmetric component of tropical waves, with the vertically upward vector representing a phase lag of 0° with phase lag increasing clockwise. Three dispersion straight lines with increasing slopes represent the equatorial Kelvin waves (derived from the shallow water equations) corresponding to three equivalent depths, 12, 25, and 50 m, respectively. (j)–(l) Composites of 20–100-day filtered OLR (W m$^{-2}$, shaded) and 850-hPa wind (m s$^{-1}$, vector) for MJO phase 5 when deep convection is the strongest over the MC and 850 hPa wind, with the reference vector (1 m s$^{-1}$) shown at the top right of each panel, and (m)– (o) 15°N–15°S averaged p-vertical velocity anomaly (Pa s$^{-1}$, shaded) and moist static energy tendency anomaly (W m$^{-2}$, contour, interval 0.003); solid, dashed, and thick-black lines represent positive, negative, and zero values, respectively. The number of days used to generate the composite is shown at the bottom right corner of each panel. (a), (d), (g), (j), and (m) are from the ERA-Interim and NOAA post-processed data (abbr. ERA-I/NOAA); (b), (e), (h), (k), and (n) are from the control experiment C–30NS; and (c), (f), (i), (l), and (o) are from the A–CTL.


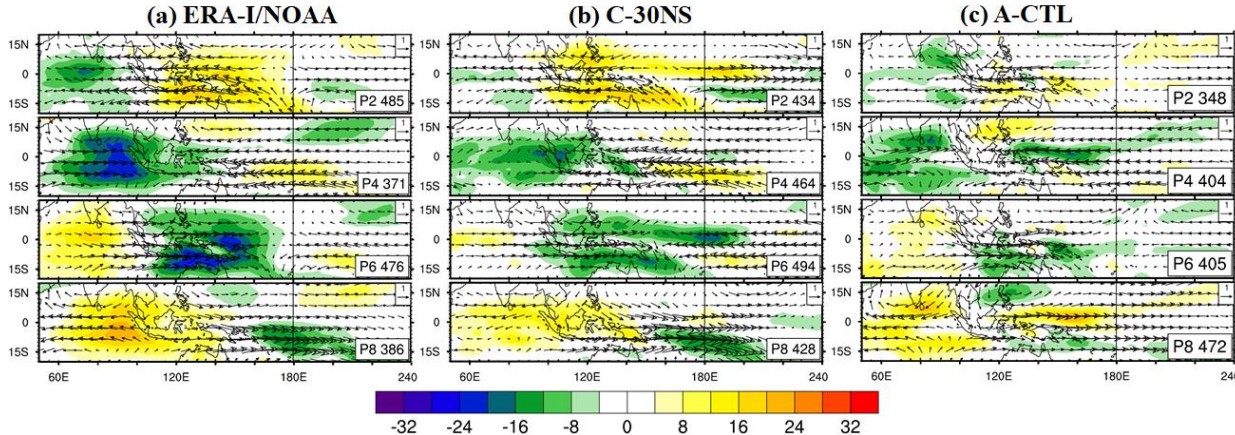



**Figure 3.** Evolution of the filtered OLR anomaly (W m$^{-2}$, shaded) and 850-hPa wind (m
s$^{-1}$, vector) at phase 2, 4, 6, and 8: (a) the ERA-I/NOAA data, (b) the control coupled
experiment C–30NS, and (c) the uncoupled experiment A–CTL. The unit of the reference
vector shown at the top right corner of each panel is m s$^{-1}$, and the number of days used
for the composite is shown at the bottom right corner of each panel.

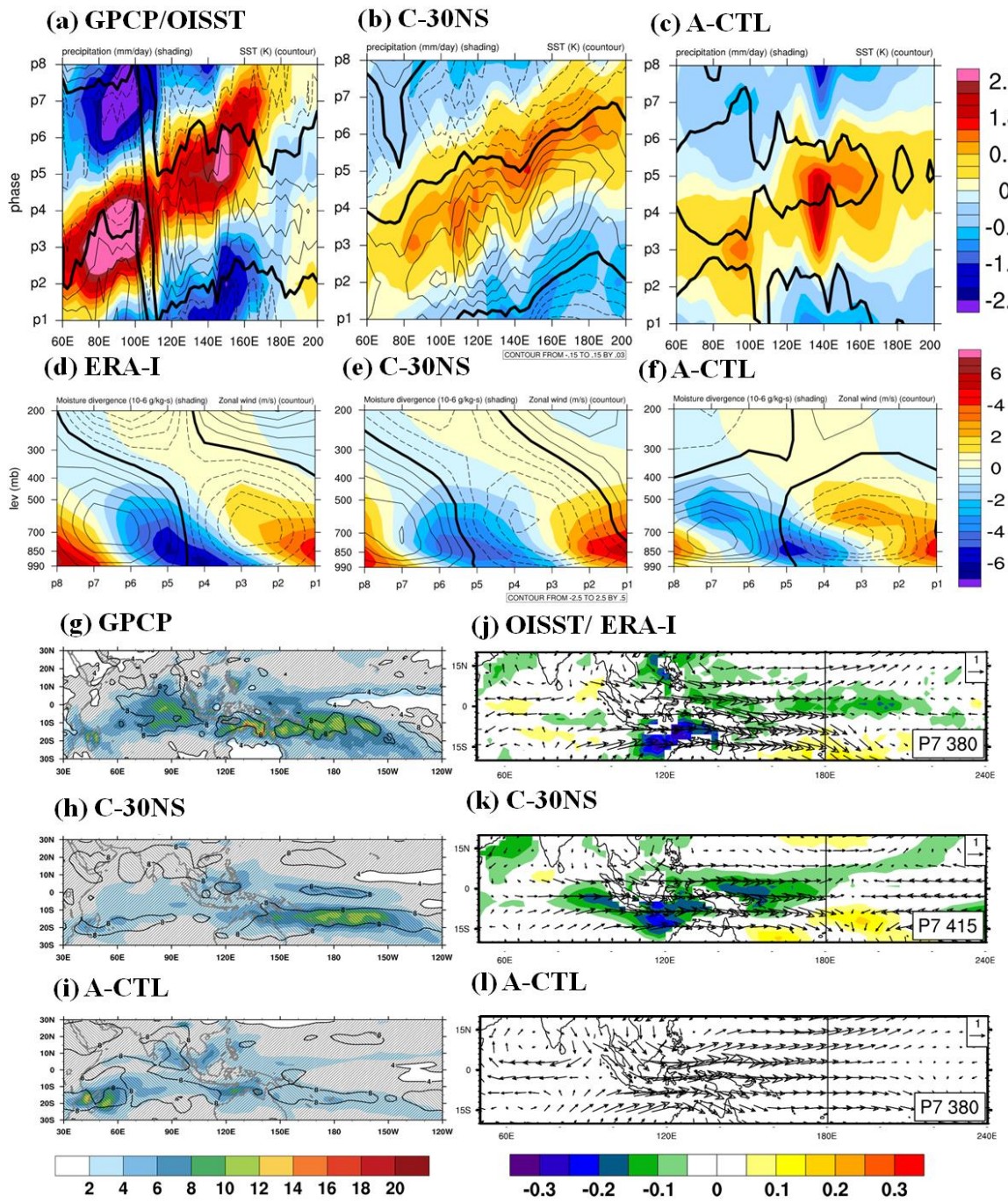

**Figure 4.** (a)–(c) Phase-longitude Hovmöller diagrams of 20–100-day filtered precipitation (mm day$^{-1}$, shaded) and SST anomaly (K, contour) averaged over 10°N–10°S from phase 1 to 8. Contour interval is 0.03; solid, dashed, and thick-black lines represent positive, negative, and zero values, respectively. (d)–(f) Phase-vertical Hovmöller diagrams of 20–100-day moisture divergence (shading, 10$^{-6}$ g kg$^{-1}$ s$^{-1}$) and zonal wind (contoured, m s$^{-1}$) averaged over 10°N–10°S, 120–150°E; solid, dashed, and thick-black curves are positive, negative, and zero values, respectively. (g)–(i) Variation of 30–60-day filtered precipitation in the eastern IO and the WP in observation (color

shading), and the ratio between intraseasonal and total variance (contoured) and (j)–(l)
composites 20–100-day filtered SST (K, shaded) and 850-hPa winds (m s$^{-1}$, vector) at
phase 7 when deep convection was the strongest over the dateline. Reference vector
shown at the top right corner of each panel. (a), (d), (g), and (j) are from the ERA-
I/NOAA data; (b), (e), (h), and (k) are from the control coupled experiment C–30NS; and
(c), (f), (i), and (l) are from the uncoupled experiment A–CTL.

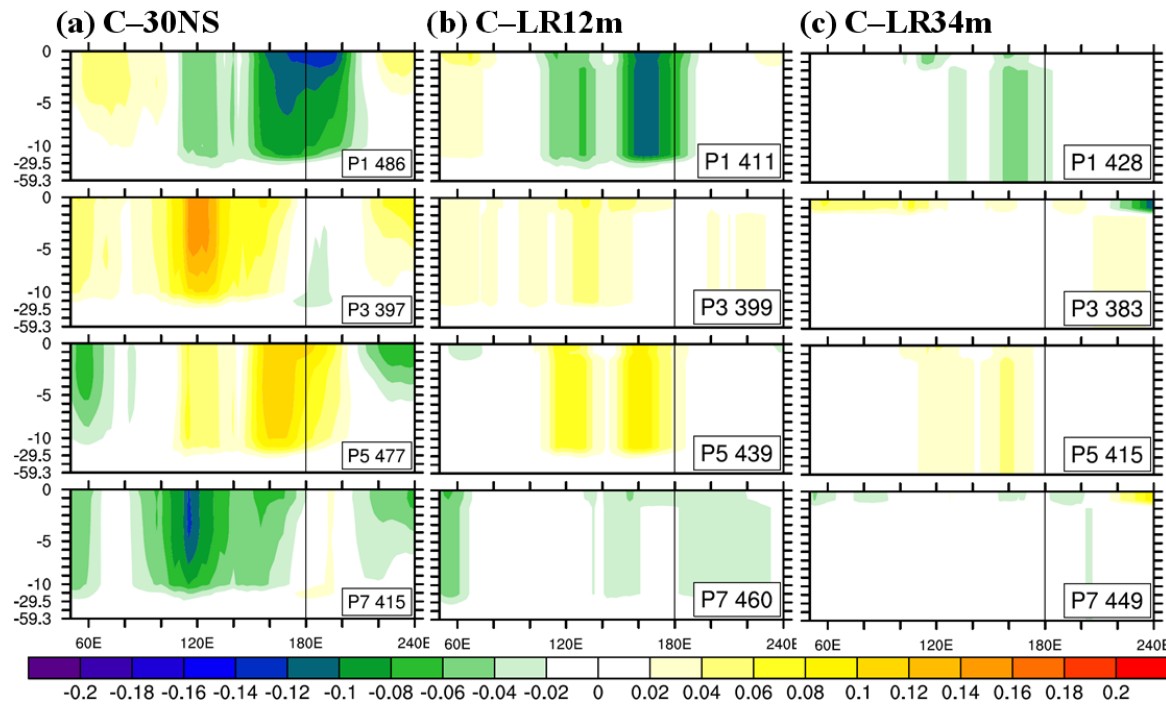



**Figure 5.** Composites of 20–100-day filtered oceanic temperature (K, shaded) between 0
and 60 m depth for MJO phase 1, 3, 5, and 7 (shown at the lower right corner of each
panel) in C–30NS, C–LR12m and C–LR34m.

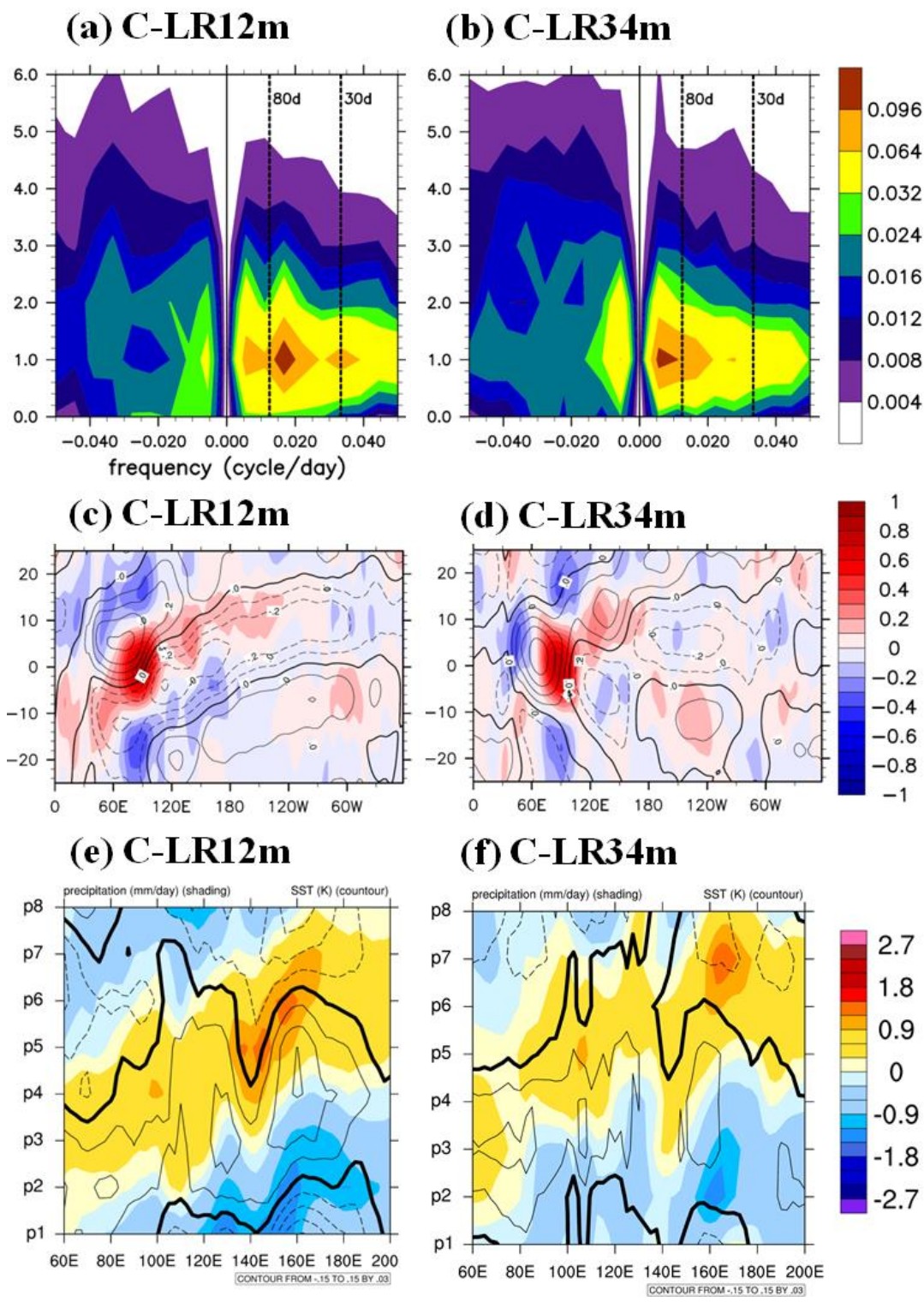


**Figure 6.** (a)–(b) Same as in Fig. 2(a) but for the C–LR12m and C–LR34m. (c)–(d) Same
as in Fig. 2(d) but for the C–LR12m and C–LR34m. (e)–(f) Same as in Fig. 4(a) but for
the C–LR12m and C–LR34m.

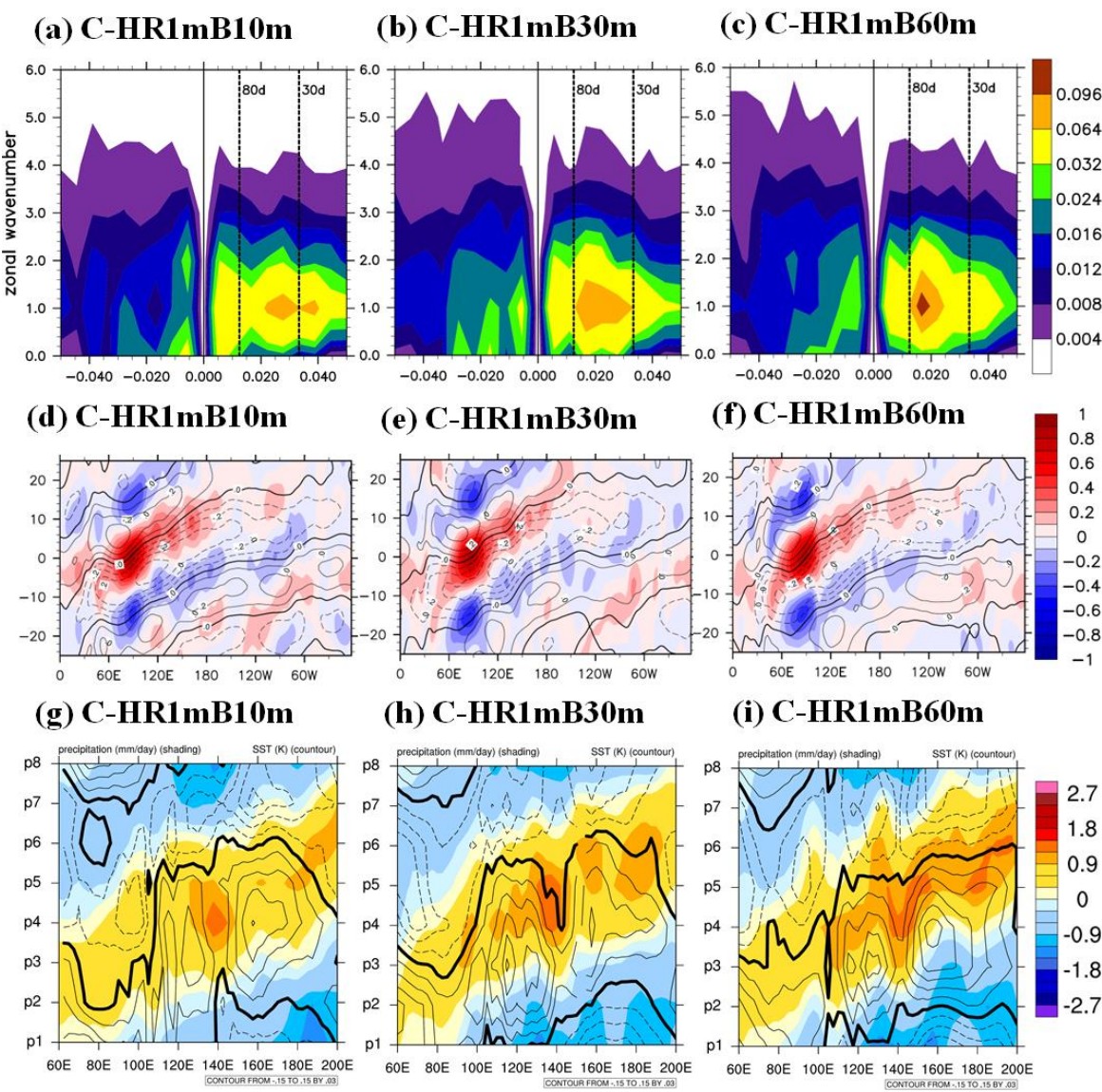



**Figure 7.** Same as in Fig. 6 but for the C–HR1mB10m, C–HR1mB30m, and C–
HR1mB60m.

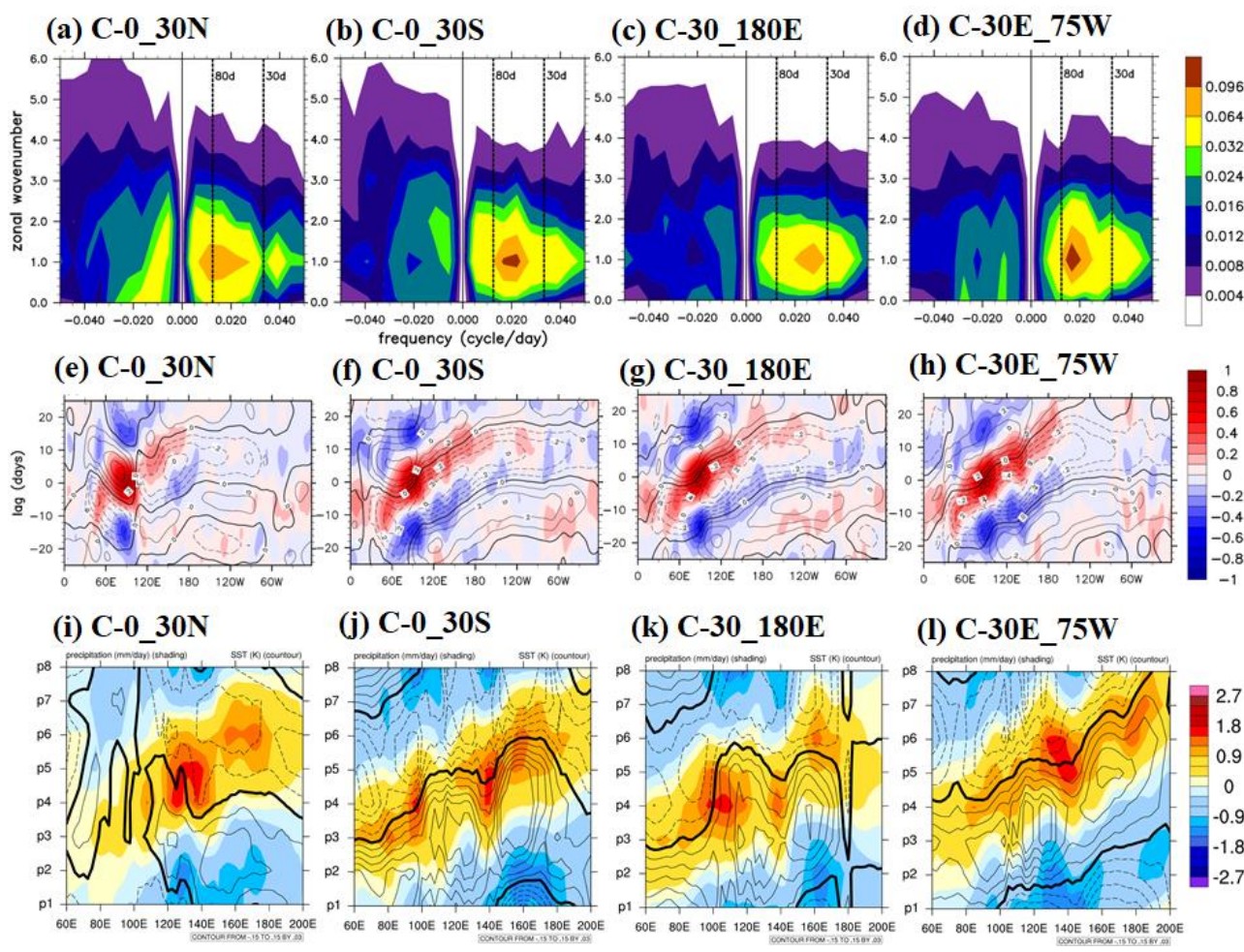



**Figure 8.** Same as in Fig. 6 but for the C–0_30N, C–0_30S, C–30_180E, and C–
30E_75W.

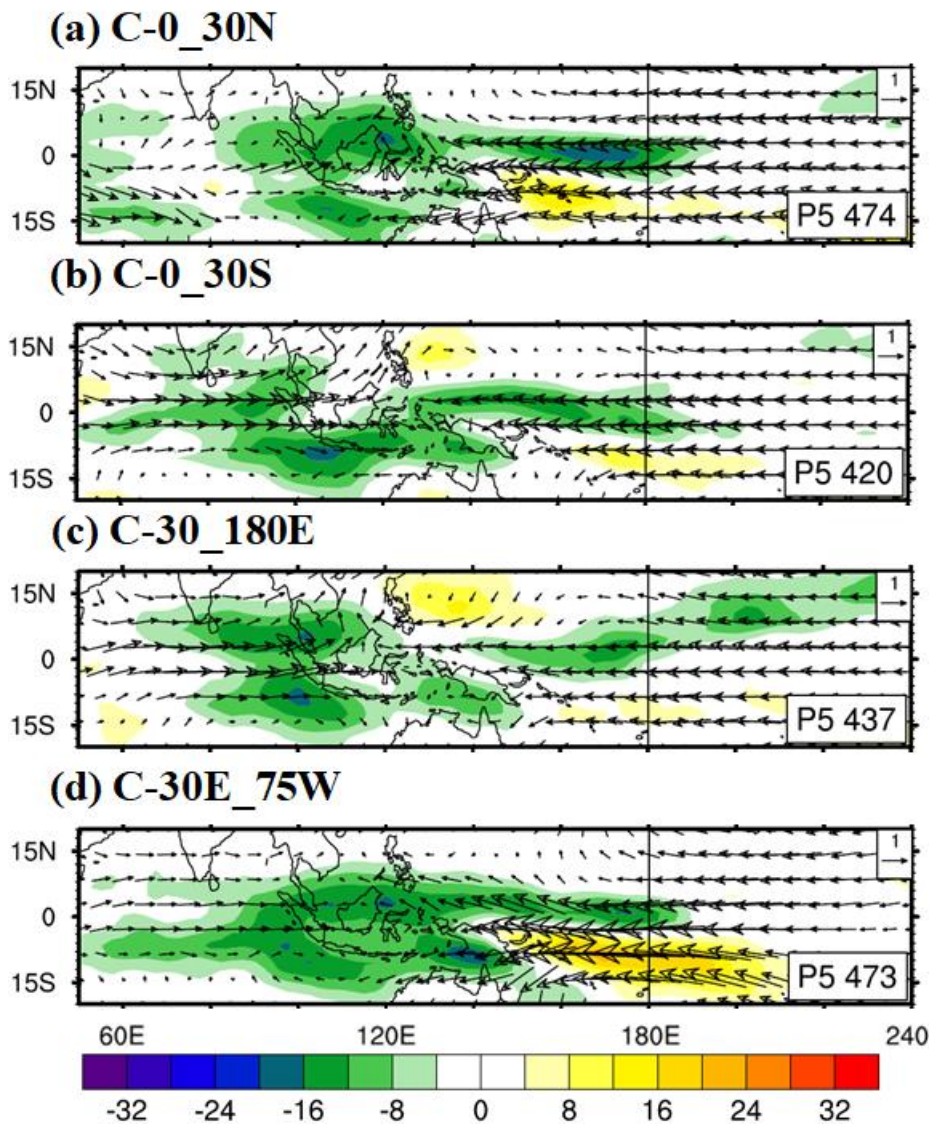

**Figure 9.** Same as in Fig. 3 but for phase 5 in the C–0_30N, C–0_30S, C–30_180E, and C–30E_75W.

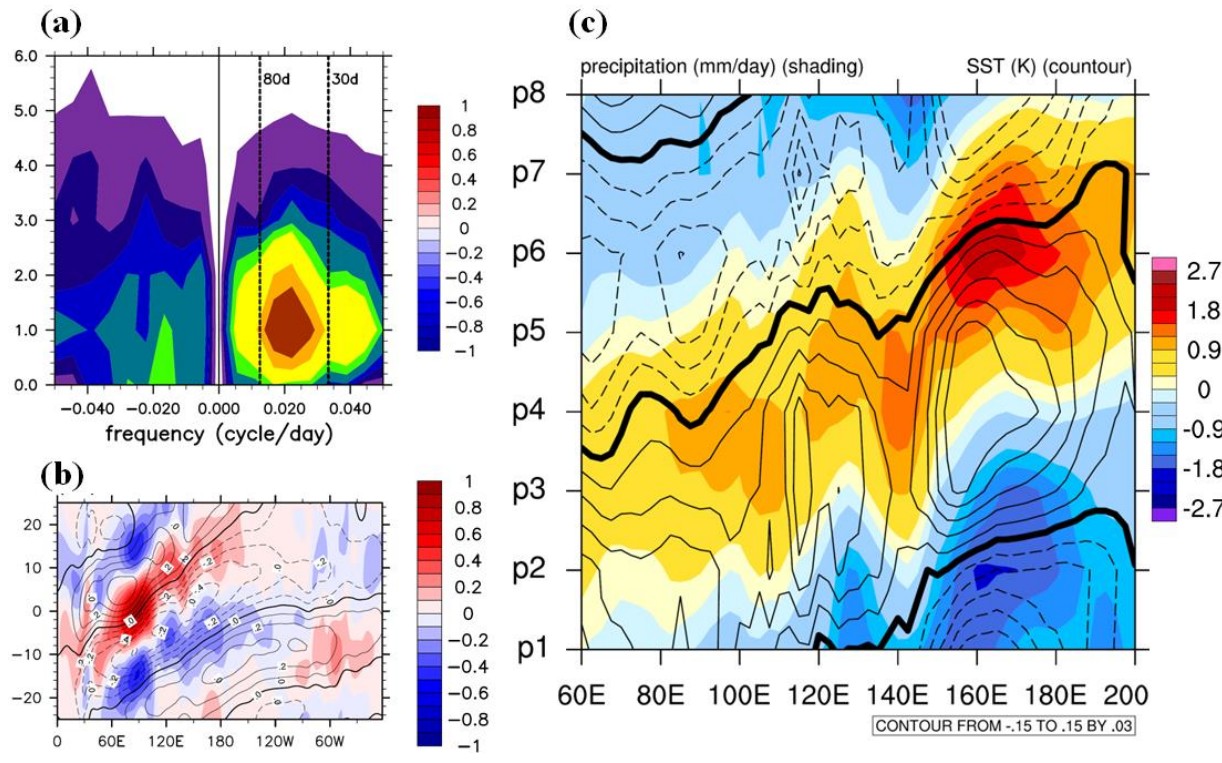

**Figure 10.** Similar as in Fig. 6 but for the C–30NS–nD.

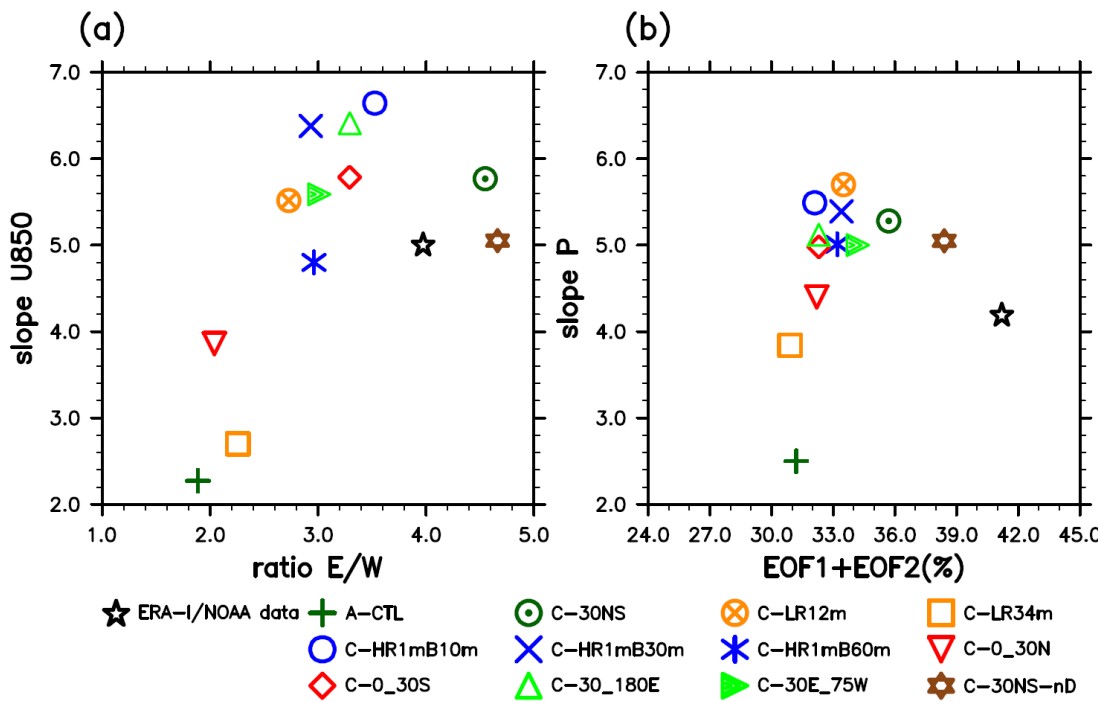

Figure 11. Scattered plots of various MJO indices in the ERA-I/NOAA data and 12 experiments: (a) power ratio of east/west propagating waves of wavenumber 1–3 of 850-hPa zonal winds (X-axis) with a 30–80-day period and eastward propagation speed of U850 anomaly (Y-axis) from the Hovmöller diagram and (b) RMM1 and RMM2 variance and eastward propagation speed of the filtered precipitation anomaly derived from the Hovmöller diagram.

**Appendix: 1-D high-resolution TKE ocean model**
The 1-D high-resolution turbulence kinetic energy ocean model SIT was used to
simulate the diurnal fluctuation of SST and surface energy fluxes. The model was well
verified against surface and subsurface observations in the South China Sea (Lan et al.,
2010) and the tropical WP (Tu and Tsuang, 2005). Variations in sea water temperature ($T$),
current ($\vec{u}$), and salinity ($S$) were determined (Gaspar et al., 1990) using the following
equations.
$$\frac{\partial T}{\partial t} = (k_h + v_h)\frac{\partial^2 T}{\partial z^2} + \frac{R_{sn}}{\rho_{w0}c_w}\frac{\partial F}{\partial z} \tag{1}$$

$$\frac{\partial \vec{u}}{\partial t} = -f\,\hat{k} \times \vec{u} + (k_m + v_m)\frac{\partial^2 \vec{u}}{\partial z^2} \tag{2}$$

$$\frac{\partial S}{\partial t} = (k_h + v_h)\frac{\partial^2 S}{\partial z^2} \tag{3}$$

where $R_{sn}$ is the net solar radiation at the surface (W m$^{-2}$), $F(z)$ is the fraction
(dimensionless) of $R_{sn}$ that penetrates to the depth $z$, and $k_h$ and $k_m$ are eddy diffusion
coefficients for heat and momentum (m$^2$ s$^{-1}$), respectively. The value of $k_h$ within the cool
skin layer and that of $k_m$ within the viscous layer were set to zero. Molecular transport is
the only mechanism for the vertical diffusion of heat and momentum in the cool skin and
viscous layer, respectively (Hasse, 1971; Grassl, 1976; Wu, 1985).The parameters $v_m$
and $v_h$ are the molecular diffusion coefficients for momentum and temperature,
respectively, $\rho_{w0}$ is the density (kg m$^{-3}$) of water, and $c_w$ is the specific heat capacity of
water at constant pressure (J kg$^{-1}$ K$^{-1}$). $S$ is salinity (‰), $\vec{u}$ is the current velocity (m s$^{-1}$),
$f$ is the Coriolis parameter (dimensionless), and $\hat{k}$ is the vertical unit vector (m s$^{-1}$).
Using the numerical solution of the surface layer ($T_0$), we disregard the time-term of
the 2-metre air temperature ($T_{2m}$), which can be considered the upper boundary of an
ocean, as well as the numerical solution of the surface long-wave radiation $T_0$ term and
aerodynamic resistance ($r_a$).

$$\frac{\partial T_0}{\partial t} = \frac{G_0}{\rho_w \cdot c_w \cdot h_e} + \frac{R_{sn}[F(z_0) - F(z_0 - d)]}{\rho_w \cdot c_w \cdot h_e} - \frac{G_{0,1}}{\rho_w \cdot c_w \cdot h_e}$$

$$= \frac{R_{ld} - R_{lu} - H - LE}{\rho_w \cdot c_w \cdot h_e} + \frac{R_{sn}[F(z_0) - F(z_0 - d)]}{\rho_w \cdot c_w \cdot h_e} - k_0 \frac{T_0 - T_1}{h_e(z_0 - z_1)}$$

$$= \frac{1}{\rho_w \cdot c_w \cdot h_e}[R_{ld} - \varepsilon\sigma T_0^4 - \frac{\rho_a c_a(T_0 - T_{2m})}{r_a} - \frac{\rho_a L_v(q^*(T_0) - q_a)}{r_a}]$$

$$+ \frac{R_{sn}[F(z_0) - F(z_0 - d)]}{\rho_w \cdot c_w \cdot h_e} - k_0 \frac{T_0 - T_1}{h_e(z_0 - z_1)} \tag{3}$$

where $G_0$ is the net flux of the ocean surface, $G_{0,1}$ is the net flux in the bottom depth of $T_0$

grid, and $K_0$ and $h_e$ are eddy diffusion coefficients and the effective thickness of $T_0$ layer,

respectively. $c_a$ is the specific heat capacity of surface air at constant pressure (J kg$^{-1}$ K$^{-1}$).

$L_v$ is the latent heat of evaporation of water $q$. We use finite difference approximation to

divide time-term into j+1 and j.

$$\frac{T_0^{j+1} - T_0^j}{\Delta t} = \frac{1}{\rho_w \cdot c_w \cdot h_e}\{R_{ld} - \varepsilon\sigma(T_0^j)^4 - \frac{\rho_a c_a}{r_a}[\beta T_0^{j+1} + (1-\beta)T_0^j - T_{2m}^j]$$

$$- \frac{\rho_a L_v(q^*(T_0) - q_a)}{r_a}\} + \frac{R_{sn}[F(z_0) - F(z_0 - d)]}{\rho_w \cdot c_w \cdot h_e} \tag{4}$$

$$- \frac{k_0}{h_e(z_0 - z_1)}[(\beta T_0^{j+1} + (1-\beta)T_0^j) - (\beta T_1^{j+1} + (1-\beta)T_1^j)]$$

$$(1 + \frac{\Delta t}{\rho_w \cdot c_w \cdot h_e}\frac{\rho_a c_a}{r_a}\beta + \frac{k_0 \cdot \Delta t}{h_e(z_0 - z_1)}\beta)T_0^{j+1} - \frac{k_0 \cdot \Delta t}{h_e(z_0 - z_1)}\beta T_1^{j+1}$$

$$= T_0^j + \frac{\Delta t}{\rho_w \cdot c_w \cdot h_e}[R_{ld} - \frac{\rho_a L_v(q^*(T_0) - q_a)}{r_a}] + \Delta t\frac{R_{sn}[F(z_0) - F(z_0 - d)]}{\rho_w \cdot c_w \cdot h_e}$$

$$+ (1 - \beta)(\frac{\Delta t}{\rho_w \cdot c_w \cdot h_e} + \frac{\rho_a c_a}{r_a} + \frac{k_0 \cdot \Delta t}{h_e(z_0 - z_1)})T_0^j - \frac{\Delta t}{\rho_w \cdot c_w \cdot h_e}\varepsilon\sigma(T_0^j)^4$$

$$+ \frac{k_0 \cdot \Delta t}{h_e(z_0 - z_1)}(1 - \beta)T_1^j \tag{5}$$

$$= T_0^j + \frac{\Delta t}{\rho_w \cdot c_w \cdot h_e}[R_{ld} - \frac{\rho_a L_v(q^*(T_0) - q_a)}{r_a} - \varepsilon\sigma(T_0^j)^4]$$

$$+ \Delta t\frac{R_{sn}[F(z_0) - F(z_0 - d)]}{\rho_w \cdot c_w \cdot h_e} - (1 - \beta)(\frac{\Delta t}{\rho_w \cdot c_w \cdot h_e} + \frac{\rho_a c_a}{r_a} + y_0)T_0^j$$

$$- (1 - \beta)y_0(T_0^j - T_1^j)$$

Since the $T_1$ is next temperature below the $T_0$, the numerical solution is based on the

average temperature of the $h_1$ layer,  $h_1 = z_0 - 0.5(z_1 + z_2)$. The parameter $\beta$ controls the
time scheme (i.e., 1 controls a backward time scheme, 0.5 controls a Crank-Nicolson
method, and 0 controls a forward time scheme).
$$\frac{\partial T_1}{\partial t} = \frac{G_0 + G_{1,2}}{\rho_w \cdot c_w \cdot h_1} + \frac{R_{sn}[F(z_0) - F(\frac{z_1 + z_2}{2})]}{\rho_w \cdot c_w \cdot h_1}$$

$$= \frac{h_e}{h_1}\frac{\partial T_0}{\partial t} + k_0 \frac{T_0 - T_1}{h_1(z_0 - z_1)} + \frac{R_{sn}[F(z_0) - F(\frac{z_1 + z_2}{2})]}{\rho_w \cdot c_w \cdot h_1} - k_1 \frac{T_1 - T_2}{h_1(z_1 - z_2)}$$

$$\frac{T_1^{j+1} - T_1^j}{\Delta t} = \frac{h_e}{h_1}\frac{T_0^{j+1} - T_0^j}{\Delta t} + k_0 \frac{[\beta T_0^{j+1} + (1-\beta)T_0^j] - [\beta T_1^{j+1} + (1-\beta)T_1^j]}{h_1(z_0 - z_1)} \quad (6)$$

$$+ \frac{R_{sn}[F(z_0 - d) - F(\frac{z_1 + z_2}{2})]}{\rho_w \cdot c_w \cdot h_1} - \frac{k_1}{h_1(z_1 - z_2)}[(\beta T_1^{j+1} + (1-\beta)T_1^j)$$

$$- (\beta T_2^{j+1} + (1-\beta)T_2^j)]$$

Specifically, the numerical solution of the next $T_2$ below the $T_1$ is not affected by the $G_0$
term, and that of the energy term is mainly affected by the $G_{1,2}$, $G_{2,3}$, and $R_{sn}$ components.
$$\frac{\partial T_2}{\partial t} = \frac{-G_{1,2} + G_{2,3}}{\rho_w \cdot c_w \cdot h_2} + \frac{R_{sn}[F(\frac{z_1 + z_2}{2}) - F(\frac{z_2 + z_3}{2})]}{\rho_w \cdot c_w \cdot h_2}$$

$$= \frac{R_{sn}[F(\frac{z_1 + z_2}{2}) - F(\frac{z_2 + z_3}{2})]}{\rho_w \cdot c_w \cdot h_2} + k_1 \frac{T_1 - T_2}{h_2(z_1 - z_2)} - k_2 \frac{T_2 - T_3}{h_2(z_2 - z_3)}$$

$$\frac{T_2^{j+1} - T_2^j}{\Delta t} = \frac{R_{sn}[F(\frac{z_1 + z_2}{2}) - F(\frac{z_2 + z_3}{2})]}{\rho_w \cdot c_w \cdot h_2} + k_1 \frac{[\beta T_1^{j+1} + (1-\beta)T_1^j] - [\beta T_2^{j+1} + (1-\beta)T_2^j]}{h_2(z_1 - z_2)} \quad (7)$$

$$- k_2 \frac{[\beta T_2^{j+1} + (1-\beta)T_2^j] - [\beta T_3^{j+1} + (1-\beta)T_3^j]}{h_2(z_1 - z_2)}$$

Similarly, the numerical solutions of layers between 3 and 41 are as follows:
$$(-\beta x_k)T_{k-1}^{j+1} + (1 + \beta \cdot x_k + \beta \cdot y_k)T_2^{j+1} - \beta \cdot y_k T_{k+1}^{j+1}$$

$$= T_k^j + \frac{\Delta t \cdot R_{sn}}{\rho_w \cdot c_w \cdot h_k}\{R_{sn}[F(\frac{z_{k-1} + z_k}{2}) - F(\frac{z_k + z_{k+1}}{2})]\} + (1-\beta)[x_k(T_{k-1}^j - T_k^j) \quad (8)$$

$$- y_2(T_k^j - T_{k+1}^j)]$$

where    $y_0 = \dfrac{\Delta t}{h_e} \dfrac{k_0}{z_0 - z_1}$

$x_k = \dfrac{\Delta t}{h_k}\left(\dfrac{k_{k-1}}{k_{k-1} - k_k}\right) \quad for \quad k = 1, n$

$y_k = \dfrac{\Delta t}{h_k}\left(\dfrac{k_k}{k_k - k_{k+1}}\right) \quad for \quad k = 1, n$

$x_g = \dfrac{\Delta t \cdot \rho_w \cdot c_w}{\rho_g \cdot c_g \sqrt{\dfrac{k_g}{w}}}\left(\dfrac{k_n}{z_n - z_g}\right)$

Finally, we get a triangular matrix for numerical solutions of 1-D high-resolution
turbulence kinetic energy ocean model SIT.

$$
\begin{bmatrix}
1+\beta\dfrac{\Delta t}{\rho_w c_w h_e}\dfrac{\rho_a c_a}{r_a}+y_0\beta & -\beta y_0 & & & \\
-\beta x_1 - \dfrac{h_e}{h_1} & 1+\beta x_1+\beta y_1 & -\beta y_1 & & \\
& & & & \\
& -\beta x_k & 1+\beta x_k+\beta y_k & -\beta y_k & \\
& & & & \\
& & & -\beta x_g & 1+\beta x_g
\end{bmatrix}
\begin{bmatrix}
T_0^{j+1} \\
T_1^{j+1} \\
\\
T_{k-1}^{j+1} \\
T_k^{j+1} \\
T_{k+1}^{j+1} \\
\\
T_n^{j+1} \\
T_g^{j+1}
\end{bmatrix}
=
$$

$$
\begin{bmatrix}
T_0^j + \dfrac{\Delta t \cdot R_{sn}[F(z_0)-F(z_0-d)]}{\rho_w c_w h_e} + \dfrac{\Delta t}{\rho_w c_w h_e}[R_{ld} - \dfrac{\rho_a L_v(q^*(T_0^j)-q_a)}{r_a} - \varepsilon\sigma(T_0^j)^4] - (1-\beta)[(\dfrac{\Delta t}{\rho_w c_w h_e}\dfrac{\rho_a c_a}{r_a}+y_0)T_0^j + y_0(T_0^j-T_1^j)] \\
T_1^j - \dfrac{h_e}{h_1}T_0^j + \dfrac{\Delta t \cdot R_{sn}}{\rho_w c_w h_1}\left[F(z_0-d) - F\left(\dfrac{z_1+z_2}{2}\right)\right] + (1-\beta)[x_1(T_0^j-T_1^j)-y_1(T_1^j-T_2^j)] \\
\\
T_k^j + \dfrac{\Delta t \cdot R_{sn}}{\rho_w c_w h_k}\left[F\left(\dfrac{z_{k-1}+z_k}{2}\right) - F\left(\dfrac{z_k+z_{k+1}}{2}\right)\right] + (1-\beta)[x_k(T_{k-1}^j-T_k^j)-y_k(T_k^j-T_{k+1}^j)] \\
\\
T_g^j + \dfrac{\Delta t R_{sn} F(z_g)}{\rho_g c_g \sqrt{\dfrac{k_g}{\omega}}} + (1-\beta)x_g(T_n^j-T_g^j)
\end{bmatrix}
$$


The eddy diffusivity for momentum $k_m$ is simulated using an eddy kinetic energy
approach based on the Prandtl–Kolmogorov hypothesis as follows:
$k_m = c_k l_k \sqrt{E}$                                           (9)
where $c_k = 0.1$ (Gaspar et al., 1990), $l_k$ is the mixing length (m), and
$E = 0.5\left(u'^2 + v'^2 + w'^2\right)$ is turbulent kinetic energy. The turbulent kinetic energy ($E$) is
determined using a 1-D equation (Mellor and Yamada, 1982) as follows:
$$\frac{\partial E}{\partial t} = \frac{\partial}{\partial z} k_m \frac{\partial E}{\partial z} + k_m \left(\frac{\partial \bar{u}}{\partial z}\right)^2 + k_h \frac{g}{\rho_w} \frac{\partial \rho_w}{\partial z} - c_\varepsilon \frac{E^{3/2}}{l_\varepsilon} \tag{10}$$

where $c_\varepsilon = 0.7$ (Gaspar et al., 1990), $g$ is the gravity (m s$^{-2}$), $\rho_w$ is the density of water
(kg m$^{-3}$), and $l_\varepsilon$ is the characteristic dissipation length (m). The mixing length ($l_k$) and
dissipation length ($l_\varepsilon$) were determined following the approach reported by Gaspar et al.
(1990). This approach is valid for determining the eddy diffusivity of both the ocean
mixed layer and surface layer.
In the SIT model setting, the specific heat of sea water is a constant (4186.84 J kg$^{-1}$
K$^{-1}$), and the Prandtl number in water is defined as the ratio of momentum diffusivity to
thermal diffusivity, which is a dimensionless number set as a constant (1.0). The
kinematic viscosity is a constant (1.14 $\times$ 10$^{-6}$ m$^2$ s$^{-1}$; Paulson and Simpson, 1981), and
the downward solar radiative flux into water with nine wavelength bands was determined
following the approach reported by Paulson and Simpson (1981). The minimum turbulent
kinetic energy is set to 10$^{-6}$ m$^2$ s$^{-2}$, and the zero displacement is set to 0.03 m.
The resolution in the upper 10.5 m is considerably fine to capture the upper-ocean
warm layer, and the thickness of the first layer below sea surface is 0.05 mm to reproduce
the ocean surface cool skin. The vertical grid within 107.8 m in C–30NS, C–LR12m and
C–LR34m as Fig. A1. Besides SST cool skin layer, C–LR12m and C–LR34m have a first
layer with grid center of -11.5 m and -33.9 m, respectively. In lowest boundary
experiment, the total vertically-gridded layers in C–HR1mB10m, C–HR1mB30m and C–
HR1mB60m are showed as Fig. A2.

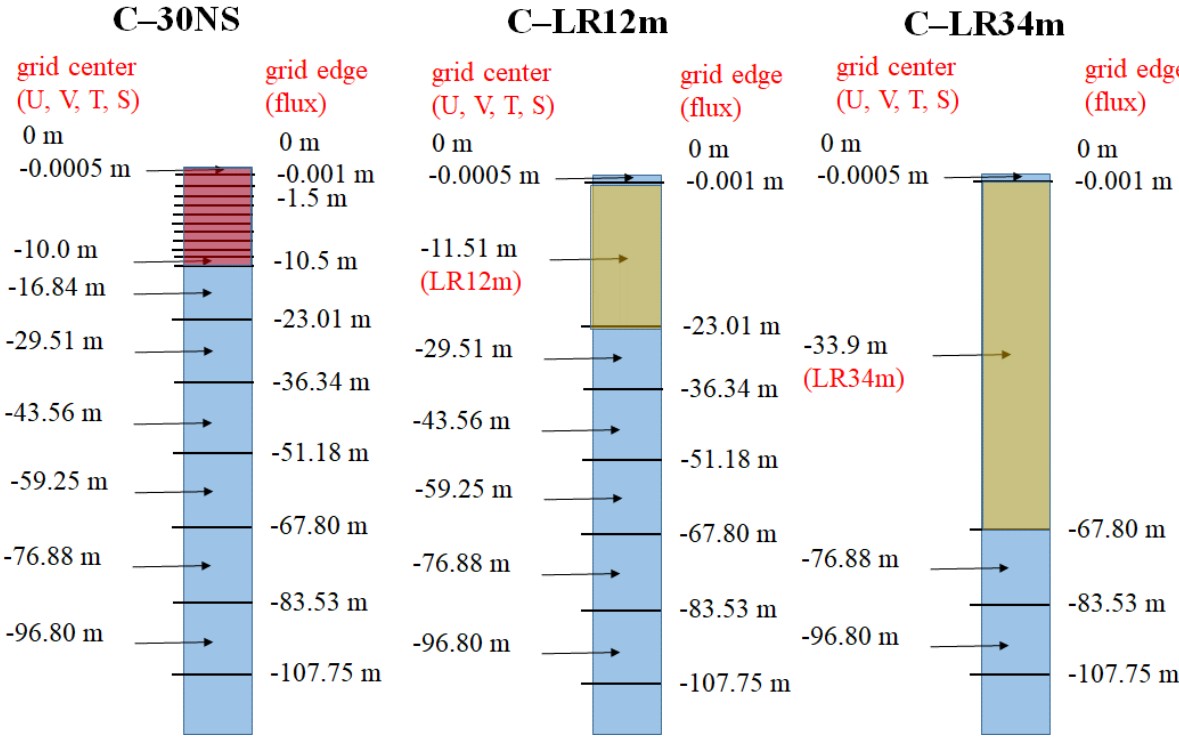

Figure A1. Diagram showing the vertical grid within 107.8 m in C–30NS, C–LR12m and
C–LR34m, the model is as thick as 107.8 meters and with several layers between surface
and model bottom. C–LR12m (31 vertical layers) and C–LR34m (28 vertical layers) have
a first layer with grid center of 12 m and 34 m, respectively, but have the same vertical
discretization as in the control experiment (C–30NS, 41 vertical layers) below the first
layer.

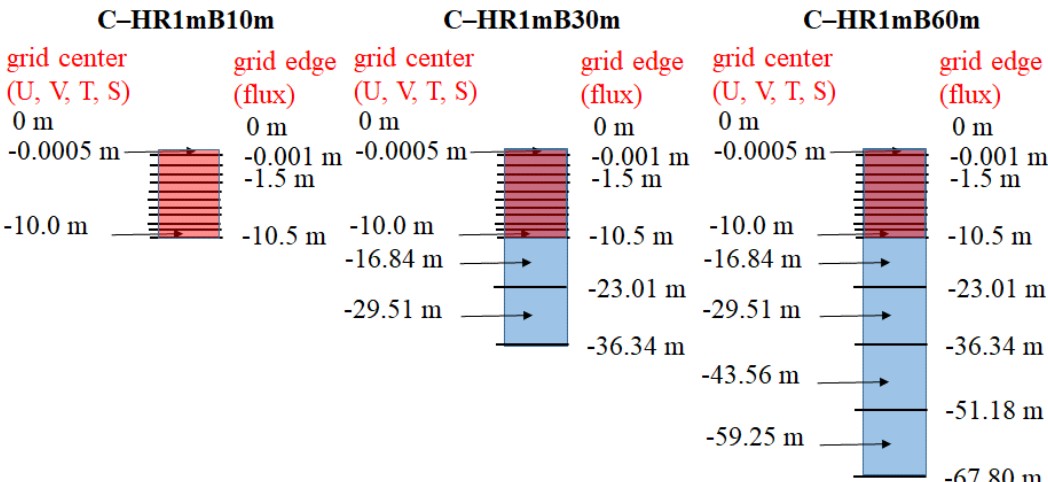


Figure A2. Diagrams showing the vertical grids in C–HR1mB10m, C–HR1mB30m and
C–HR1mB60m. The model bottoms are 10, 30, and 60 m, respectively, unless the seabed
is shallower than the above depth.

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
