# Peer review of "Embedding a One-column Ocean Model (SIT 1.06) in the Community Atmosphere Model 5.3 (CAM5.3; CAM5–SIT v1.0) to Improve Madden–Julian Oscillation Simulation in Boreal Winter"

_Geoscientific Model Development, 2021_

## Author Comment (AC1)

Dear Editors of GMD and dear Reviewers:

We greatly appreciate reviewer's insightful and helpful comments regarding our manuscript. The manuscript has been revised based on reviewer's comments. Below are the point-by-point replies to reviewer's comments and concerns, whereas our corresponding revisions in the manuscript (version R1) are identified by colored text. Specifically, red text indicates changes made in response to the suggestions from Reviewer #1, blue text demonstrates changes made according to Reviewer #2, and green text shows changes made to better clarify model descriptions in a clear, concise, and well-structured way. Moreover, we revised the manuscript carefully to ensure that it is grammatically and typographically error-free and hopefully meets the high quality standards of GMD.

Sincerely,
Yung-Yao Lan, Huang-Hsiung Hsu, Wan-Ling Tseng, and Li-Chiang Jiang

Anonymous Referee #1
The reviewer comments are formatted in italics and the authors response to the comments are formatted in bold.
Notation *RC1.P#* represents Reviewers Comment. Paragraph Number

> *RC1.general comment 1. This manuscript focuses on the development of a global coupled model on forecasting MJOs. The propagation of MJOs along the equator can significantly affect the precipitation in many regions, so the relevant model works have been devoted by many previous studies. I appreciate the authors' efforts for continuously improving the model forecast on this multi-scale weather system. Unfortunately, one thing I am trying to find in this manuscript is their unique contributions to the broad society. According to the title, it seems like the authors feeling confident in the usage of a 1-D SIT model for predicting MJOs. At the end of Introduction, the authors barely mention their motivation is to "examine how air–sea coupling can improve MJO simulation, especially that of the eastward propagation that has been poorly simulated in many climate models". Because many global coupled models use the 3-D ocean models, the connection between the title (1-D SIT model) and motivation (effect of air-sea coupling on MJO propagation) is unclear. Are the authors trying to convince readers the effect of 1-D model enough for the forecast? Or is there anything special inside the SIT model? The importance of air-sea coupling should have been extensively emphasized and agreed by many studies, and I do not think any ongoing research still trying to use a global model without ocean parts. Repeating the work may be meaningless. I believe their motivation needs to be rewritten.*

**Response:**

**Thank you for your comment. We did not attempt to argue that the effect of**

**1-D model enough for the forecast or simulation of the MJO; instead, we**

**demonstrate that a 1-D model with high vertical resolution in the first 10 meters**

**could have significant improvement. At the end, we suggested that using extra**

**fine vertical resolution in the first few tens of meters of 3-D ocean model could**

**further improve the simulation of the MJO. The improvement due to high**

**resolution had been demonstrated using ECHAM5 (Tseng et al. 2014). This**

**study demonstrated the same effect in CAM5 and suggested that the**

**improvement is not model dependence. By coupling the 1-D SIT model to an**

**AGCM different from Tseng et al. (2014), this study confirms the scientific**

**reproducibility for the improvement of MJO simulation in modeling science.**

**We further explored the dependence of the improvement on various factors**

such as coupling depth, frequency and domain that have not been explored in previous studies, and we considered our results valuable insights for the MJO

simulations. We have revised the introduction and summary following the discussion above to state more clearly the motivation and contribution of this study.

> RC1.general comment 2. On the other hand, because the authors introduce some
> models unable to simulate the MJO propagation reliably, I believe one of their
> expected results is to improve the motion of MJOs (also mentioned in the
> motivation). However, it seems like the authors do not summarize how much
> improvements can be seen in their results, or which factors can affect the simulation
> the most. Because there are some interesting experiments inside this manuscript,
> such as the coupling regions, I do not think it should be rejected at this moment.
> However, the structure and quality of the manuscript are very poor. It is very close
> to my standard for rejection (too many things to be fixed). I only list some problems
> below, not all. I recommend a major revision for this work in this review.

Response:

Thanks for your suggestion. We summarized specifically in the original (and revised) manuscript what are the better settings and important factors for

MJO simulations. We did not attempt to quantify the degree of improvement because it is likely model dependent. Nevertheless, the improvement is evident in many presented figures, e.g., the summarized figure (Figure 10 in revised manuscript) shown in the Summary. The findings are as follows.

(1) Better resolving the fine structure of the upper-ocean temperature and therefore the air–sea interaction led to more realistic intraseasonal variability in both SST and atmospheric circulation.

(2) An adequate thickness of the oceanic mixed layer is required to simulate a delayed response of the upper ocean to atmospheric forcing and lower- frequency fluctuation.

**(3) Coupling the tropical eastern Pacific, in addition to the tropical IO and the**

**tropical WP, can enhance the MJO and facilitate the further eastward**

**propagation of the MJO to the dateline.**

**(4) Coupling the southern tropical ocean, instead of the norther tropical ocean, is**

**essential for simulating a realistic MJO.**

**(5) Stronger MJO variability can be obtained without considering the diurnal**

**cycle in coupling.**

**In general, upper-ocean vertical resolution and coupling with the southern**

**tropical would be of relative importance compared to other factors for the**

**eastward propagation of the MJO.**

> *RC1.P1 I do not think conducting an experiment for studying the difference between A-CTL and C-30NS is needed. In my point of view, we do not need another paper talking about the importance of coupling the upper ocean in the global models. In other words, please simplify the description in section 4.1. All you need is to show your coupled model sufficient for simulating the MJOs.*

**Response:**

**The purpose of the comparison between A–CTL and C–30NS was not just**

**to demonstrate again that air–sea coupling could improvement MJO simulation.**

**It also served as the basis for the evaluation of sensitivity experiments that tested**

**the key ingredients for the improvement, in addition to showing that significant**

**improvement in MJO simulation can be achieved by simply coupling a**

**numerically efficient 1-D ocean model. For this purpose, the C–30NS experiment**

**served as a control coupled experiment is essential. We therefore prefer to retain**

**this experiment and relevant discussion, and hope for reviewer's understanding.**

> *RC1.P2 I am super uncomfortable in the description of the ERA-interim results as the "observation". It is impossible to measure the global wind at 850 hPa directly. Besides, the precipitation data looks like a post-processed product constituted by many satellite measurements. It happens to the OISST as well.*

**Response:**

**Thank you for the suggestion. We modified the manuscript to mention**

**directly the name of data used for comparison, instead of referring them as**

**observation. Please see Page 11, lines 244, 247 and 260, Page 12, lines 272, 274**

**and 280 as well as section 3 with red text in the revised manuscript.**

> *RC1.P3 I think you need to reconsider your structure in the main text. There are some unnecessary and redundant materials that can be moved to the appendix or supplemental material. For example, you do not adjust the coefficients in the 1-D TKE closure scheme. Why do you need to describe the full equations? I also don't care about the numbers of depths from lines 207 to 212 (yes, your units are wrong).*

**Response:**

**The comments are well taken. We have removed the background**

**information about SIT and the units are corrected. Thank you for the reminder.**

**Please see Page 7, lines 159-161 and Page 8, lines 162-180.**

> *RC1.P4 You do not need section 3, because people like me already forget the details when we are reading sections since 4.2. Please reorganize the structure.*

**Response:**

**Thank you for the suggestion. We feel a brief discussion of experiment setups**

**could be useful for completeness and the readers. Content of Section 3 is now**

**moved to Section 2.3. The essence of each experiment was briefly mentioned**

**again in other sections when relevant results were presented. Detailed**

**information of each experiment is also presented in a table and in supplementary**

**material.**

[Figure]

**Fig. RC1.1 Schematic diagram of a series of 30-year numerical experiments.**

**Table 1. List of experiments**

| Section | Category | Experiments | Description |
|---|---|---|---|
| 3.1 | Coupled or uncoupled | A–CTL | Standalone CAM5.3 forced by forced by the monthly mean Hadley Centre SST dataset version 1 climatology |
| | | C–30NS (the control coupled experiment) | CAM5.3 coupled with SIT over the tropical domain (30°S–30°N), with 41 layers of finest vertical resolution (up to the seabed) and diurnal cycle; the frequency of CAM5 being exchanged with CPL is 48 times per day |
| 3.2 | Upper-ocean vertical resolution | C–LR12m | The first ocean vertical level starts at 11.5 m with 31 layers (beside SST and cool skin layer are 11.5 m, 29.5 m and 43.6 m up to the seabed) |
| | | C–LR34m | The first ocean vertical level starts at 33.9 m with 28 layers (beside SST and cool skin layer are 33.9 m, 76.9 m and 96.8 m up to the seabed) |
| 3.3 | Lowest boundary of SIT | C–HR1mB10m | The lowest boundary of SIT has a depth of 10 m (model depth between 0 m and 10 m) |
| | | C–HR1mB30m | The lowest boundary of SIT has a depth of 30 m (model depth between 0 m and 30 m) |
| | | C–HR1mB60m | The lowest boundary of SIT has a depth of 60 m (model depth between 0 m and 60 m) |
| 3.4 | Regional coupling domain in latitude | C–0_30N | Coupled in the tropical northern hemisphere (0°N–30°N, 0°E–360°E) |
| | | C–0_30S | Coupled in the tropical southern hemisphere (0°S–30°S, 0°E–360°E) |
| | Regional coupling domain in longitude | C–30_180E | Coupled in the Indo-Pacific (30°S–30°N, 30°E–180°E) |
| | | C–30E_75W | Coupled over the Indian Ocean and Pacific Ocean (30°S–30°N, 30°E–75°W) |
| 3.5 | Absence of the diurnal cycle | C–30NS–nD | Absence of the diurnal cycle in C–30NS; the CAM5.3 daily atmospheric mean of surface wind, temperature, total precipitation, net surface heat flux, u-stress and v-stress over water trigger the SIT and daily mean SST feedback to atmosphere; the frequency of CAM5 is exchanged with CPL 48 times per day |

**Experiment abbreviations: "A" means standalone AGCM simulation. "C"**

**means the CAM5.3 coupled to the SIT model.**

*RC1.P5 I do not think that section 4.2 is discussing the vertical resolution… It is more like the thickness of the first layer. A lot of information is missing here. For example, what is your surface mixed layer depth? If the surface mixed layer depth is less than 30 m or 10 m, what do you do for C-LR34m C-LR12m? Are you trying to test the effect of a slab model in your global coupled model?*

**Response:**

**At the first sight, it may seem as reviewer suggested "more like the thickness**

**of the first layer". Although we did not conduct different vertical resolutions**

**within the first 10.5 meters, a comparison between three experiments did suggest**

**that the extra fine resolution in the first 10 meters contribute markedly to the**

**improvement. With a 41-layer vertical discretization in SIT model in the control**

**experiment, 12 layers are located above 10.5 m and 6 layers are located between**

**10.5 m and 107.8 m. High vertical resolution is needed to catch detailed temporal**

**variation of upper ocean temperature. To test the effect of vertical resolution, we**

**conducted C–LR12m and C–LR34m without vertical discretization in the first**

**layer (Figure RC1.2) to explore the impacts of fine vertical resolution on MJO**

**simulation. This comparison showed that the simulated MJO became more**

**realistic with increasing the upper-ocean vertical resolution. This result has an**

**important implication for the further development of fully coupled GCM that**

**often has the first oceanic layer as thick as 10 meters (e.g., POP2).**

**The SIT is not a simple slab model that usually has just one layer. As shown**

**in Figure RC1.2, the model is as thick as 107.8 meters and with several layers**

**between surface and model bottom. C–LR12m and C–LR34m have a first layer**

**with grid center at 12m and 34m, respectively, but have the same vertical**

**discretization as in the control experiment (C–30NS). We apologize for the**

**confusion. Figure RC1.2 is now included in supplementary material. Readers can**

**better understand the experiment setups.**

**SIT vertical grid mixing processes are based on eddy and molecular**

**diffusivity for heat and momentum. The numerical treatments of C–LR12m (31**

**vertical layers) and C–LR34m (28 vertical layers) would still be computed from 0**

**m to seabed if the mixed layer depth was less than 30 m or 10 m.**

[Figure]

**Fig. RC1.2 Diagram showing the vertical grid within 107.8 m in C–30NS, C–**

**LR12m and C–LR34m.**

*RC1.P6 What do you mean "ocean bottom" at line 476? Is it seafloor?*

**Response:**

**Thank you for the question. "Ocean bottom" is misleading. It should be the bottom**

**of the SIT as shown in Fig. RC1.3. Their ocean model bottoms are 10, 30, and 60 m,**

**respectively, unless the seabed is shallower than the above depth. For example, if**

**the seafloor of ocean grid is deeper than 67.8 m, this ocean grid of C–HR1mB60m**

**would be computed from 0 m to 59.3 m depth. IF the seafloor is 52 m depth in one**

**of C–HR1mB60m ocean grid, this grid would only be computed from 0 m to 43.6 m**

**depth. We have change "ocean bottom" to "ocean model bottom" in the**

**manuscript. Please see Page 9, lines 211-213 and Page 19, line 464.**

[Figure]

**Fig. RC1.3 Diagram showing the totally vertical grids in C–HR1mB10m, C–**

**HR1mB30m and C–HR1mB60m.**

> *RC1.P7 Rewrite section 4.6. I cannot understand which fluxes you are using.*

**Response:**

**Heat fluxes here were sensible and latent fluxes that were calculated based**

**on simulated winds, moisture, and temperature. We have modified the text**

**accordingly in revised manuscript. Thank you for the reminder. Please see Page**

**3, line 50 and Page 22, lines 539-542.**

> *RC1.P8 I cannot understand why the runs are 30 yr? What are the initial conditions*
> *of atmosphere and ocean? Is the forcing the same as the values in the real world*
> *from 1990-2020?*

**Response:**

**A 30-year period is commonly used to define a current climate by the WMO**

**and IPCC (2013) and has been a common length adopted in climate simulations**

**to produce stable statistics. It is natural for us to adopt the same simulation**

**strategy.**

**All simulations were driven by the same emission and annual cycle of SST**

**for 30 years. The strategy is to evaluate the ability of model under the same**

conditions without considering interannual variation. This approach has been widely adopted in many studies (Delworth et al., 2006; Haertel et al., 2020;

Subramanian et al., 2011; Tseng et al., 2014; Wang et al., 2005). Based on the atmosphere component of the Community Earth System Model version 1.2.2

(CESM1.2.2) framework development, all experiments of CAM5–SIT were conducted under the F_2000_CAM5 component set that provides the near- equilibrium climate responses. The sea surface temperature (SST, HadSST1)

used to force the model was the climatological monthly means SST averaged over

1982-2001. The monthly SST was linearly interpolated to daily SST fluctuation that forced the model. The SST in air–sea coupling region was recalculated by

SIT during the simulation, while the prescribed annual cycle of SST was used in the areas outside the coupling region.

Atmospheric initial conditions and other external forcing such as $CO_2$, ozone, and aerosol representing the climate around year 2000 were taken from the default setting of F_2000_CAM5 component set that has been commonly used in present-day simulation using CAM5 (e.g., He et al., 2017). Initital conditions were not needed for the SST that was prescribed as lower boundary condition in the experiments. This information is now included in the revised manuscript.

References:
Delworth, T. L., et al.: GFDL's CM2 global coupled climate models.
Part 1: Formulation and simulation characteristics. J. Climate,
19, 643–674, https://doi.org/10.1175/JCLI3629.1, 2006.

Haertel, P.: Prospects for Erratic and Intensifying Madden-Julian
Oscillations. Climate, 8, 24, https://doi.org/10.3390/cli8020024,
2020.

He, S., Yang, S. and Li, Z.: Influence of Latent Heating over the
Asian and Western Pacific Monsoon Region on Sahel Summer
Rainfall, Sci Rep 7, 7680, https://doi.org/10.1038/s41598-017-
07971-6, 2017.

IPCC: Annex III: Glossary [Planton, S. (ed.)]. In: Climate Change 2013: The Physical Science Basis. Contribution of Working Group I to the Fifth Assessment Report of the Intergovernmental Panel on Climate Change [Stocker, T.F., D. Qin, G.-K. Plattner, M. Tignor, S.K. Allen, J. Boschung, A. Nauels, Y. Xia, V. Bex and P.M. Midgley (eds.)]. Cambridge University Press, Cambridge, United Kingdom and New York, NY, USA. 2013.

Subramanian, A. C., Jochum, M., Miller, A. J., Murtugudde, R., Neale, R. B., and Waliser, D. E.: The Madden–Julian oscillation in CCSM4, J. Climate, 24, 6261–6282, https://doi.org/10.1175/JCLI-D-11-00031.1, 2011.

Tseng, W.-L., Tsuang, B.-J., Keenlyside, N. S., Hsu, H.-H. and Tu, C.-Y.: Resolving the upper-ocean warm layer improves the simulation of the Madden-Julian oscillation, Clim. Dynam., 44, 1487–1503, https://doi.org/10.1007/s00382-014-2315-1, 2014.

Wang, S. Saha, Pan, H. L., Nadiga, S. and White, G.: Simulation of ENSO in the new NCEP Coupled Forecast System Model (CFS03). Mon. Wea. Rev., 133, 1574–1593, https://doi.org/10.1175/MWR2936.1, 2005.

---

## Author Comment (AC2)

Dear Editors of GMD and dear Reviewers:

Thank you for the positive feedback and for thoroughly reading the manuscript with constructive comments. Appropriated changes, suggested by the Reviewer #2, has been introduced to the manuscript. The following is a point-by-point response to the reviewer's concerns, whereas our corresponding revisions in the manuscript (version R1) are identified by colored text. Specifically, red text indicates changes made in response to the suggestions from Reviewer #1, blue text demonstrates changes made according to Reviewer #2, and green text shows changes made to better clarify model descriptions in a clear, concise, and well-structured way. Moreover, we revised the manuscript carefully to ensure that it is grammatically and typographically error-free and hopefully meets the high quality standards of GMD.

Sincerely,
Yung-Yao Lan, Huang-Hsiung Hsu, Wan-Ling Tseng, and Li-Chiang Jiang

Anonymous Referee #2
The reviewer comments are formatted in italics and the authors response to the comments are formatted in bold.
Notation *RC2.P#* represents Reviewers Comment. Paragraph Number

> *RC2.P1 When describing model results, I would suggest to use "present tense" instead of "past tense" throughout the paper.*

**Response:**

**Thanks for your kind reminders. In the revised manuscript, we describe the model results in the present tense.**

> *RC2.P2 Line 37: move "in the year 2011" after "Dynamics of the MJO"?*

**Response:**

**The modifications are part of "an overview of findings from a multi-nation**

**field campaign called Dynamics of MJO/Cooperative Indian Ocean Experiment**

**on Intraseasonal Variability in the Year 2011 (DYNAMO/CINDY2011)" in the**

**revised manuscript. Please see Page 3, lines 36-39.**

* * *
*RC2.P3 Line 68: may delete "and climate models"*
* * *
**Response:**

**The revised manuscript removes the wordiness from this sentence. Please**

**see Page 4, line 71.**
* * *
*RC2.P4 Line 109: may change to "regarding the effect of air-sea coupling on the MJO"?*
* * *
**Response:**

**To make reading easier, we corrected this statement as reviewer's suggestion.**

**Please see Page 6, lines 112-113.**

* * *
*RC2.P5 Line 273-274: Are U850 anomalies not averaged over 10N-10S, instead of just on the equator?*
* * *
**Response:**

**This was indeed an unclear statement in the original manuscript. These**

**modifications are described as follows: "Figure 2d–f show the time evolution of**

**precipitation and U850 anomalies in Hovmöller diagrams, which represent**

**lagged correlation coefficients between the precipitation averaged over 10°S–**

**5°N, 75–100°E and the precipitation and U850 averaged over 10°N–10°S on**

**intraseasonal timescales". Please see Page 11, lines 251-255.**

* * *
*RC2.P6 In general, figure quality can be improved (many look blur with detals difficult to identify), and some figures can be a bit enlarged.*
* * *
**Response:**

**Thank you for the suggestions. Figure quality has been improved and size has**

**been enlarged.**

* * *
*RC2.P7 Line 305: the "observed" MJO characteristics*
* * *
**Response:**

**In response to the suggestion by another reviewer that ERA-Interim**

**reanalysis and NOAA post-processed satellite data (ERA-I/NOAA) should not be**

**referred to as "observation", we have modified the description to "In summary,**

**C–30NS produce coherent and energetic patterns in the eastward-propagating**

**intraseasonal fluctuations of U850 and OLR in the tropical IO and WP that are**

**generally consistent with the MJO characteristics derived from ERA-I and**

**NOAA OLR". Please see Page 12, lines 283-288.**

* * *
*RC2.P8 Line 467: in the first few meters "below the surface" allows ....?*
* * *
**Response:**

**Thank you for the suggestion. It has been modified to "This result confirms**

**the finding reported by Tseng et al. (2014) that a higher vertical resolution in the**

**upper few meters below the sea surface allows for a faster air–sea interaction,**

**thus resulting in a more realistic simulation of the MJO". Please see Page 19,**

**lines 454-456.**

* * *
*RC2.P9 Line 556: I didn't see faster MJO propagation when the diurnal coupling is turned off based on Fig. 9b. If compared to Fig. 5a, seems to me the MJO propagation speed is even faster in the C-30NS run with diurnal coupling. This is also related to the following comments on Fig. 10. Generally, I don't see significant differences in MJO simulations between the no-diurnal coupling experiment and the control experiment.*
* * *
**Response:**

**Thank you for the comment. Fig. 9b should be compared with Fig. 2e**

**instead of Fig. 5b. A comparison by eye inspection is not easy to see the**

**difference. Propagation speeds estimated based on the Hovmöller diagrams of**

**U850 and precipitation are shown in Fig. 10. For U850, the MJO with diurnal**

**cycle (marked by target sign) is faster than the one with no diurnal cycle**

**(marked by Star of David sign). The difference is more evident for U850. We**

**agree that the difference is very small for precipitation. The statement is**

**modified as above in revised manuscript. Please see Page 22, lines 547-550.**

RC2.P10 Fig. 10: It would be better provide more details on how the U850 and P slopes are determined, e.g., based on which longitude bands. Also the colors for "C-30NS-nD" are not consistent between the figure and legend.

**Response:**

**In the revised manuscript, we corrected the conflicting colors between the**

**figures and the legend (Fig. RC2.1). Based on the maximum precipitation**

**anomaly and zero values of U850 (indicating deep convection region),**

**propagation speeds of U850 and precipitation are calculated from Hovmöller**

**diagram on intraseasonal timescales between 60°E and 150°W. Please see Page**

**24, lines 585-588.**

[Figure]

**Fig. RC2.1 Scattered plots of various MJO indices in the ERA-I/NOAA data and 12 experiments: (a) power ratio of east/west propagating waves of wavenumber 1–3 of 850-hPa zonal winds (X-axis) with a 30–80-day period and eastward propagation speed of U850 anomaly (Y-axis) from the Hovmöller diagram and (b) RMM1 and RMM2 variance and eastward propagation speed of the filtered precipitation anomaly derived from the Hovmöller diagram.**

**References:**
**Tseng, W.-L., Tsuang, B.-J., Keenlyside, N. S., Hsu, H.-H. and Tu, C.-Y.: Resolving the upper-ocean warm layer improves the simulation of the Madden-Julian oscillation, Clim. Dynam., 44, 1487–1503, https://doi.org/10.1007/s00382-014-2315-1, 2014.**

---

## Referee Report (RR1)

This is my second time for reviewing the manuscript drafted by the authors (Lan et al.). Compared with the previous version, I would like to compliment their efforts on reorganizing the structure. At least for now, it is more readable to me for understanding the messages they want to deliver. The points in their response also answer most of my concerns in the previous version. To be honest, the current status of the manuscript is publishable, after some minor revisions are done. However, I hope the authors can spend some efforts in revising or adding more details about the ocean model parts. It may benefit more readers for understanding the importance for each experiment mentioned in section 2.3. If both the editor and authors think these comments are unnecessary, it is ok to just move on to the next step.

1. Section 2.3 now clearly lists the five experiments finished in this manuscript. However, it will be useful to describe the reasons behind each experiment more. For example,
    A. Section 3.1. describes that the C-30NS is aimed to compared with A-CTL. I hope some descriptions can be added either in the introduction or section 3.1. for explaining why coupling in the tropical region is more important than that in the high latitudes (yeah, people can guess MJO as a tropical atmosphere system, but it can still be helpful)
    B. The reason behind the experiment in section 3.2 is about the effect of fine vertical resolution in the ocean model. However, it is very interesting to see that the authors try to demonstrate it by making the thickness of the layer (the one below the SST layer) up to 10 or 30 m. I hope the authors can give more physical explanations on the reasons for doing it. I can expect less temperature changes if this layer is thicker, but why testing it? Normally, it may be done by changing vertical resolution near sea surface. Because the vertical resolution in the upper 10 m of C-30NS is ~1 m, I may decrease the vertical resolution in the upper 10 m, instead of setting a thick near-surface layer.
    C. Section 3.3 is the experiment I still cannot understand after the revision… Line 462 wants to study how thick a vertically-gridded ocean mixed layer. It makes me expect the authors will artificially average the temperature or salinity structure near the sea surface. Line 464 then mentions "the ocean model (SIT) bottom at 10, 30, and 60 m, which included the top 12, 14, and 16 levels". From Table 1, the authors describe it as the thickness of the ocean model is 10, 30 and 60 m, respectively. It seems like a confliction between line 462 and 464 to me. To me, artificially mixing the near-surface

layer is more reasonable, because the heat during the air-sea interaction can be downward transported to more than 60-m depth via turbulent mixing. Setting the bottom of ocean model less than a certain number is to force the heat to be trapped. It will for sure affect the SST, but may not be consistent with the authors' purpose in discussing the effect of surface mixed layer.

    D. I don't have any questions for the sections 3.4 and 3.5.

2. Because I do not expect I will review this manuscript once again, and the manuscript may be published after this revision, I suggest the authors pay extra efforts in checking the grammar or errors within sentences. For example,

- Line 64: may, in turn, "yield"
- Line 142: which "considered" the (tense needs to be consistent in each paragraph)
- Line 225: "air-sea"

---

## Author Response (AR2)

Dear editor and reviewer,

We appreciate editor's positive comments and reviewer's insightful remarks. The manuscript has been revised based on reviewer's comments and the grammatical and typographical errors have been corrected to meet the high-quality standard of GMD. In response to reviewer's comment, one figure is added in the revised manuscript as Fig. 5. Below are the point-by-point replies to reviewer's comments.

Sincerely,
Yung-Yao Lan, Huang-Hsiung Hsu, Wan-Ling Tseng, and Li-Chiang Jiang

Anonymous Referee #1
Reviewer's comments are formatted in italics and the authors' response are formatted in bold.

*RC1.general comment.*

*This is my second time for reviewing the manuscript drafted by the authors (Lan et al.). Compared with the previous version, I would like to complement their efforts on reorganizing the structure. At least for now, it is more readable to me for understanding the messages they want to deliver. The points in their response also answer most of my concerns in the previous version. To be honest, the current status of the manuscript is publishable, after some minor revisions are done. However, I hope the authors can spend some efforts in revising or adding more details about the ocean model parts. It may benefit more readers for understanding the importance for each experiment mentioned in section 2.3. If both the editor and authors think these comments are unnecessary, it is ok to just move on to the next step.*

**Response:**

**We agree with reviewer that adding this discussion would be helpful to readers in understanding the significance of each experiment. A detailed explanation of the ocean model is provided in the appendix.**

> *Section 2.3 now clearly lists the five experiments finished in this manuscript.*
> *However, it will be useful to describe the reasons behind each experiment more.*
> *For example,*
>
> *A. Section 3.1. describes that the C-30NS is aimed to compared with A-CTL. I hope*
> *some descriptions can be added either in the introduction or section 3.1. for*
> *explaining why coupling in the tropical region is more important than that in the*
> *high latitudes (yeah, people can guess MJO as a tropical atmosphere system, but*
> *it can still be helpful).*

**Response:**

**Thank you for the suggestion. Reasons are added to explain the purpose of**

**each experiment design.**

> *B. The reason behind the experiment in section 3.2 is about the effect of fine vertical*
> *resolution in the ocean model. However, it is very interesting to see that the*
> *authors try to demonstrate it by making the thickness of the layer (the one below*
> *the SST layer) up to 10 or 30 m. I hope the authors can give more physical*
> *explanations on the reasons for doing it. I can expect less temperature changes if*
> *this layer is thicker, but why testing it? Normally, it may be done by changing*
> *vertical resolution near sea surface. Because the vertical resolution in the upper*
> *10 m of C-30NS is ~1 m, I may decrease the vertical resolution in the upper 10*
> *m, instead of setting a thick near-surface layer.*

**Response:**

**We agree with reviewer's point that testing different vertical resolutions in**

**top 10 meters would be another way demonstrating the necessity of high vertical**

**resolution for better MJO simulation. In this study, we did not attempt to**

**identify the optimal vertical resolution for MJO simulation, instead we chose to**

**demonstrate the significant improvement that a fine vertical resolution can**

**achieve compared to the coarse resolution (e.g., tens of meters) that is often**

**adopted in slab ocean model. Through the comparison we also demonstrated the**

**crucial role of air-sea interaction, which can only be well simulated with fine**

**vertical resolution, in shaping the characteristics of MJO. Reviewer's suggestion**

**is well taken. We'll test the idea in a following experiment and hopefully to**

present the results in another report. Our hunch is that the incremental decrease in vertical resolution likely worsens the simulation gradually, but not as dramatically as demonstrated in the C–LR12m and C–LR34m simulations. We added Fig. RC1 in the revised manuscript to demonstrate the dramatic changes in vertical profile of ocean temperature between the fine and coarse vertical resolution simulations. Amplitude of ocean temperature decreases dramatically in coarse resolution simulations. In addition, there is a clear vertical stratification of ocean surface temperatures in C–30NS, whereas C–LR12m and

C–LR34m are well mixed without obvious stratification. This demonstrates the necessity of fine vertical gridding for resolving the fast fluctuation of ocean temperature when interacting with the atmosphere.

[Figure]

Fig. RC1 Composites of 20–100-day filtered oceanic temperature (K, shaded)

between 0 and 60 m depth for MJO phase 1, 3, 5, and 7, shown at the top right of each panel in C–30NS, C–LR12m and C–LR34m.

> *C. Section 3.3 is the experiment I still cannot understand after the revision… Line 462 wants to study how thick a vertically-gridded ocean mixed layer. It makes me expect the authors will artificially average the temperature or salinity structure near the sea surface. Line 464 then mentions "the ocean model (SIT) bottom at 10, 30, and 60 m, which included the top 12, 14, and 16 levels". From Table 1, the authors describe it as the thickness of the ocean model is 10, 30 and 60 m, respectively. It seems like a confliction between line 462 and 464 to me. To me, artificially mixing the near-surface layer is more reasonable, because the heat during the air-sea interaction can be downward transported to more than 60-m depth via turbulent mixing. Setting the bottom of ocean model less than a certain number is to force the heat to be trapped. It will for sure affect the SST, but may not be consistent with the authors' purpose in discussing the effect of surface mixed layer.*

**Response:**

   We apologize for not well explaining the essence of the experiment reported in Section 3.3. In this set of experiment, all experiments retained same vertical resolution (e.g., 1 meter in the first top 10 meters of the ocean) but with various ocean bottom (i.e., 10, 30, and 60 meters in the experiment). The purpose is to demonstrate how the total ocean heat content, which depends on the total depth of the ocean, can affect the MJO. Considering two models with the same vertical resolution, the model with thinner ocean (e.g., 10 meter) would interact as efficiently as another model with thicker ocean (e.g., 60m) but with much less heat to release to or to absorb from the atmosphere. The former would have less impact on the atmosphere than the latter. Perhaps, the word "thickness" or "mixed layer" confuses the reviewer to think the model has a well-mixed upper ocean. In fact, the ocean in the model could still be stratified if in stable condition (e.g., under calm weather condition). We make this point more clearly in the revised manuscript. The corresponding text has been rewritten. We avoided using those terms such as mixed layer that would confuse readers.

> *D. I don't have any questions for the sections 3.4 and 3.5.*

**Response:**

**Thank you.**

> *Because I do not expect I will review this manuscript once again, and the manuscript may be published after this revision, I suggest the authors pay extra efforts in checking the grammar or errors within sentences. For example,*
>
> - *Line 64: may, in turn, "yield"*
> - *Line 142: which "considered" the (tense needs to be consistent in each paragraph)*
> - *Line225: "air-sea"*

**Response:**

**Thank you for the suggestion. We have carefully revised the manuscript and**

**corrected grammar errors. The errors spotted by the reviewer have been**

**corrected. Please see lines 65, lines 144 and lines 243.**